# GENERALIZATION PROPERTIES OF RETRIEVAL-BASED MODELS

## ABSTRACT

Many modern high-performing machine learning models such as GPT-3 primarily rely on scaling up models, e.g., transformer networks. Simultaneously, a parallel line of work aims to improve the model performance by augmenting an input instance with other (labeled) instances during inference. Examples of such augmentations include task-specific prompts and similar examples retrieved from the training data by a nonparametric component. Remarkably, retrieval-based methods have enjoyed success on a wide range of problems, ranging from standard natural language processing and vision tasks to protein folding, as demonstrated by many recent efforts, including WebGPT and AlphaFold. Despite growing literature showcasing the promise of these models, the theoretical underpinning for such models remains underexplored. In this paper, we present a formal treatment of retrieval-based models to characterize their generalization ability. In particular, we focus on two classes of retrieval-based classification approaches: First, we analyze a local learning framework that employs an explicit *local empirical risk minimization* based on retrieved examples for each input instance. Interestingly, we show that breaking down the underlying learning task into local sub-tasks enables the model to employ a *low complexity* parametric component to ensure good overall accuracy. The second class of retrieval-based approaches we explore learns a global model using kernel methods to directly map an input instance and retrieved examples to a prediction, without explicitly solving a local learning task.

## 1 INTRODUCTION

As our world is complex, we need expressive machine learning models to make high accuracy predictions on real world problems. There are multiple ways to increase expressiveness of a machine learning model. A popular way is to homogeneously scale the size of a parametric model, such as neural networks, which has been behind many recent high-performance models such as GPT-3 (Brown et al., 2020) and ViT (Dosovitskiy et al., 2021). Their performance (accuracy) exhibits a monotonic behavior with increasing model size, as demonstrated by "scaling laws" (Kaplan et al., 2020). Such large models, however, have their own limitations, including high computation cost, catastrophic forgeting (hard to adapt to changing data), lack of provenance, and explanability. Classical instance-based models Fix & Hodges (1989), on the other hand, offer many desirable properties by design — efficient data structures, incremental learning (easy addition and deletion of knowledge), and some provenance for its prediction based on the nearest neighbors w.r.t. the input. However, these models often suffer from weaker empirical performance as compared to deep parametric models.

Increasingly, a middle ground combining the two paradigms and retaining the best of both worlds is becoming popular across various domains, ranging from natural language (Das et al., 2021; Wang et al., 2022; Liu et al., 2022; Izacard et al., 2022), to vision (Liu et al., 2015; 2019; Iscen et al., 2022; Long et al., 2022), to reinforcement learning (Blundell et al., 2016; Pritzel et al., 2017; Ritter et al., 2020) , to even protein structure predictions (Cramer, 2021) . In such approaches, given a test input, one first retrieves relevant entries from a data index and then processes the retrieved entries along with the test input to make the final predictions using a machine learning model. This process is visualized in Figure 1b. For example, in semantic parsing, models that augment a parametric seq2seq model with similar examples have not only outperformed much larger models but also are more robust to changes in data (Das et al., 2021).

While classical learning setups (cf. Figure 1a) have been studied extensively over decades, even basic properties and trade-offs pertaining to retrieval-based models (cf. Figure 1b), despite their

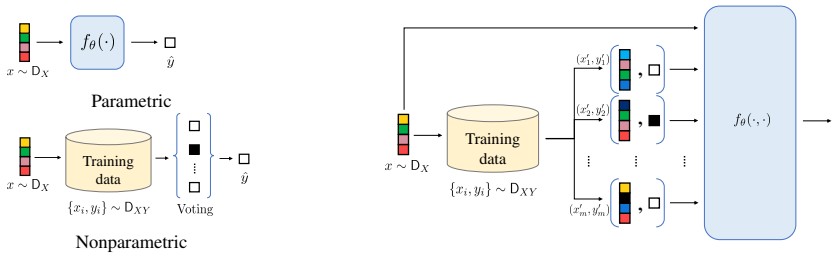

(a) Classical learning setups        (b) Modern retrieval-based setup

Figure 1: An illustration of a retrieval-based classification model. Given an input instance $x$, similar to an instance-based model, it retrieves similar (labeled) examples $\mathcal{R}^x = \{(x'_j, y'_j)\}_j$ from training data. Subsequently, it processes (potentially via a nonparametric method) input instance along with the retrieved examples to make the final prediction $\hat{y} = f(x, \mathcal{R}^x)$.

aforementioned remarkable successes, remain highly under-explored. Most of the existing efforts on retrieval-based machine learning models solely focus on developing end-to-end *domain-specific* models, without identifying the key dataset properties or structures that are critical in realizing performance gains by such models. Furthermore, at first glance, due to the highly dependent nature of an input and the associated retrieved set, direct application of existing statistical learning techniques does not appear as straightforward. This prompts the natural question: *What should be the right theoretical framework that can help rigorously showcase the value of the retrieved set in ensuring superior performance of modern retrieval-based models?*

In this paper, we take the first step towards answering this question, while focusing on the classification setting (Sec. 2.1). We begin with the hypothesis that the model might be using the retrieved set to do local learning implicitly and then adapt its predictions to the neighborhood of the test point. This idea is inspired from Bottou & Vapnik (1992). Such local learning is potentially beneficial in cases where the underlying task has a local structure, where a much simpler function class suffices to explain the data in a given local neighborhood but overall the data can be complex (formally defined in Sec. 2.2). For instance looking at a few answers at Stackoverflow even if not for same problem may help us solve our issue much faster than understanding the whole system. We try to formally show this effect.

We begin by analyzing an explicit local learning algorithm: For each test input, (1) we retrieve a few training examples located in the vicinity of the test input, (2) train a local model by performing empirical risk minimization (ERM) with only these retrieved examples – *local ERM*; and (3) apply the resulting local model to make prediction on the test input. For the aforementioned retrieval-based local ERM, we derive finite sample generalization bounds that highlight a trade-off between the complexity of the underlying function class and size of neighborhood where local structure of the data distribution holds in Sec. 3. Under this assumption of local regularity, we show that by using a much simpler function class for the local model, we can achieve a similar loss/error to that of a complex global model (Thm. 3.4). Thus, we show that breaking down the underlying learning task into local sub-tasks enables the model to employ a *low complexity* parametric component to ensure good overall accuracy. Note that the local ERM setup is reminiscent of semiparametric polynomial regression (Fan & Gijbels, 2018) in statistics, which is a special case of our setup. However, the semiparametric polynomial regression have been only analyzed asymptotically under mean squared error loss (Ruppert & Wand, 1994) and its treatment under a more general loss is unexplored.

We acknowledge that such local learning cannot be the complete picture behind the effectiveness of retrieval-based models. As noted in Zakai & Ritov (2008), there always exists a model with global component that is more "preferable" to a local-only model. In Sec. 3.2, we extend local ERM to a two-stage setup: First learn a global representation using entire dateset, and then utilize the representation at the test time while solving the local ERM as previously defined. This enables the local learning to benefit from good quality global representations, especially in sparse data regions.

Finally, we move beyond explicit local learning to a setting that resembles more closely the empirically successful systems such as REINA, WebGPT, and AlphaFold: A model that directly learns to predict from the input instance and associated retrieved similar examples end-to-end. Towards this, we take a preliminary step in Sec. 4 by studying a novel formulation of classification over an extended feature space (to account for the retrieved examples) by using kernel methods (Deshmukh et al., 2019).

To summarize, our main contributions include: 1) Setting up a formal framework for classification under local regularity; 2) Finite sample analysis of explicit local learning framework; 3) Extending the analysis to incorporate a globally learnt model; and 4) Providing the first rigorous treatment of an end-to-end retrieval-based models to understand its generalization by using kernel-based learning.

## 2 PROBLEM SETUP

We first provide a brief background on (multiclass) classification along with necessary notations. Subsequently, we discuss the problem setup considered in this paper, which deals with designing retrieval-based classification models for the data distributions with local regularity.

### 2.1 MULTICLASS CLASSIFICATION

In this work, we restrict ourselves to (multi-class) classification setting, with access to $n$ training examples $\mathcal{S} = \{(x_i, y_i)\}_{i \in [n]} \subset \mathcal{X} \times \mathcal{Y}$, sampled i.i.d. from the data distribution $\mathsf{D} := \mathsf{D}_{X,Y}$. Given $\mathcal{S}$, one is interested in learning a classifier $h : \mathcal{X} \to \mathcal{Y}$ that minimizes miss-classification error. It is common to define a classifier via a scorer $f : x \mapsto \big(f_1(x), \ldots, f_{|\mathcal{Y}|}(x)\big) \in \mathbb{R}^{|\mathcal{Y}|}$ that assigns a score to each class in $\mathcal{Y}$ for an instance $x$. For a scorer $f$, the corresponding classifier takes the form: $h_f(x) = \arg\max_{y \in \mathcal{Y}} f_y(x)$. Furthermore, we define the *margin* of $f$ at a given label $y \in \mathcal{Y}$ as

$$\gamma_f(x, y) = f_y(x) - \max_{y' \neq y} f_{y'}(x). \tag{1}$$

Let $\mathbb{P}_{\mathsf{D}}(A) := \mathbb{E}_{(X,Y) \sim \mathsf{D}} \big[\mathbb{1}_{\{A\}}\big]$ for any random variable $A$. Given $\mathcal{S}$ and a set of scorers $\mathcal{F} \subseteq \{f : \mathcal{X} \to \mathbb{R}^{|\mathcal{Y}|}\}$, learning a model implies finding a scorer in $\mathcal{F}$ that minimizes miss-classification error:

$$f^* = \arg\min_{f \in \mathcal{F}} \mathbb{P}_{\mathsf{D}}(h_f(X) \neq Y). \tag{2}$$

One typically employs a surrogate loss (Bartlett et al., 2006) $\ell$ for the miss-classification loss $\mathbb{1}_{\{h_f(X) \neq Y\}}$ and aims minimize the associated risk:

$$R_\ell(f) = \mathbb{E}_{(X,Y) \sim \mathsf{D}}\big[\ell\big(f(X), Y\big)\big]. \tag{3}$$

Since the underlying data distribution $\mathsf{D}$ is only accessible via examples in $\mathcal{S}$, one learns a good scorer by minimizing the (global) *empirical* risk over a large function class $\mathcal{F}^{\mathrm{global}}$ as follows:

$$\hat{f} = \arg\min_{f \in \mathcal{F}^{\mathrm{global}}} \widehat{R}_\ell(f) := \frac{1}{n} \sum_{i \in [n]} \ell\big(f(x_i), y_i\big). \tag{4}$$

### 2.2 DATA DISTRIBUTIONS WITH LOCAL REGULARITY

In this work, we assume that the underlying data distribution $\mathsf{D}$ follows a local-regularity structure, where a much simpler (parametric) function class suffices to explain the data in each local neighborhood. Formally, for $x \in \mathcal{X}$ and $r > 0$, we define $\mathcal{B}^{x,r} := \{x' \in \mathcal{X} : \mathrm{d}(x, x') \leq r\}$, an $r$-radius ball around $x$, w.r.t. a metric $\mathrm{d} : \mathcal{X} \times \mathcal{X} \to \mathbb{R}$. Let $\mathsf{D}^{x,r}$ be the data distribution restricted to $\mathcal{B}^{x,r}$, i.e.,

$$\mathsf{D}^{x,r}(A) = \mathsf{D}(A)/\mathsf{D}\big(\mathcal{B}^{x,r} \times \mathcal{Y}\big) \quad A \subseteq \mathcal{B}^{x,r} \times \mathcal{Y}. \tag{5}$$

Now, the *local regularity condition* of the data distribution ensures that, for each $x \in \mathcal{X}$, there exists a low-complexity function class $\mathcal{F}^x$, with $|\mathcal{F}^x| \ll |\mathcal{F}^{\mathrm{global}}|$, that approximates the Bayes optimal (w.r.t. $\mathcal{F}^{\mathrm{global}}$) for the *local classification problem* defined by $\mathsf{D}^{x,r}$. That is, for a given $\varepsilon_{\mathcal{X}} > 0$, we have[1]

$$\min_{f \in \mathcal{F}^x} \mathbb{E}_{\mathsf{D}^{x,r}}[\ell(f(X), Y)] \leq \min_{f \in \mathcal{F}^{\mathrm{global}}} \mathbb{E}_{\mathsf{D}^{x,r}}[\ell(f(X), Y)] + \varepsilon_{\mathcal{X}}, \quad \forall x \in \mathcal{X}. \tag{6}$$

As an example, if $\mathcal{F}^{\mathrm{global}}$ is linear in $\mathbb{R}^d$ (possibly dense) with bounded norm $\tau$, then $\mathcal{F}^x$ can be a simpler function class such as linear in $\mathbb{R}^d$ with sparsity $k \ll d$ and with bounded norm $\tau_x \leq \tau$.

### 2.3 RETRIEVAL-BASED CLASSIFICATION MODEL

This work focuses on retrieval-based methods that can leverage the aforementioned local regularity structure of the data distribution. In particular, we focus on two such approaches:

---

[1]As stated, we require the local-regularity condition to hold for each $x$. This can be relaxed to hold with high probability with increased complexity of exposition.

**Local empirical risk minimization.** Given a (test) instance $x$, the *local* empirical risk minimization (ERM) approach first retrieves a neighboring set $\mathcal{R}^x = \{(x'_j, y'_j)\} \subseteq \mathcal{S}$. Subsequently, it identifies a (local) scorer $\hat{f}^x$ from a 'simple' function class $\mathcal{F}^{\mathrm{loc}} \subset \{f : \mathcal{X} \to \mathbb{R}^{|\mathcal{Y}|}\}$ as follows:

$$\hat{f}^x = \arg\min_{f \in \mathcal{F}^{\mathrm{loc}}} \hat{R}^x_\ell(f); \quad \hat{R}^x_\ell(f) := \frac{1}{|\mathcal{R}^x|} \sum_{(x', y') \in \mathcal{R}^x} \ell(f(x'), y'). \tag{7}$$

Here, $\mathcal{R}^x$ corresponds to the samples in $\mathcal{S}$ that belong to $\mathcal{B}^{x,r}$; hence, it follows the distribution $\mathsf{D}^{x,r}$. We assume there exists $N(r, \delta)$ such that for any $r \geq 0$, and $\delta > 0$,

$$\mathbb{P}_{(X,Y) \sim \mathsf{D}}\big[|\mathcal{R}^X| < N(r, \delta)\big] \leq \delta, \text{ and } \mathbb{P}_{(X,Y) \sim \mathsf{D}}\big[|\mathcal{R}^X| = 0\big] = 0. \tag{8}$$

Note that the local ERM approach requires solving a local learning task for each test instance. Such a local learning algorithms was introduced in Bottou & Vapnik (1992). Another point worth mentioning here is that (7) employs the same function class $\mathcal{F}^{\mathrm{loc}}$ for each $x$, whereas the local regularity assumption (cf. (6)) allows for an instance dependent function class $\mathcal{F}^x$. We consider $\mathcal{F}^{\mathrm{loc}}$ that approximates $\cup_{x \in \mathcal{X}} \mathcal{F}^x$ closely. In particular, we assume that, for some $\varepsilon_{\mathrm{loc}} > 0$, we have

$$\min_{f \in \mathcal{F}^{\mathrm{loc}}} \mathbb{E}_{\mathsf{D}^{x,r}}[\ell(f(X), Y)] \leq \min_{f \in \mathcal{F}^x} \mathbb{E}_{\mathsf{D}^{x,r}}[\ell(f(X), Y)] + \varepsilon_{\mathrm{loc}}, \quad \forall\, x \in \mathcal{X}. \tag{9}$$

Continuing with the example following (6), where $\mathcal{F}^x$ is linear with sparsity $k \ll d$ and bounded norm $\tau_x$, one can take $\mathcal{F}^{\mathrm{loc}}$ to be linear with the same sparsity $k$ and bounded norm $\tau' < \sup_{x \in \mathcal{X}} \tau_x$.

**Classification with extended feature space.** Another approach to leverage the retrieved neighboring labeled instances during classification is to directly learn a scorer that maps $x \times \mathcal{R}^x \in \mathcal{X} \times (\mathcal{X} \times \mathcal{Y})^\star$ to per-class scores. One can learn such a scorer over *extended* feature space $\mathcal{X} \times (\mathcal{X} \times \mathcal{Y})^\star$ as follows:

$$\hat{f}^{\mathrm{ex}} = \arg\min_{f \in \mathcal{F}^{\mathrm{ex}}} \hat{R}^{\mathrm{ex}}_\ell(f); \quad \hat{R}^{\mathrm{ex}}_\ell(f) := \frac{1}{n} \sum_{i \in [n]} \ell\big(f(x_i, \mathcal{R}^{x_i}), y_i\big), \tag{10}$$

where $\mathcal{F}^{\mathrm{ex}} \subset \big\{f : \mathcal{X} \times (\mathcal{X} \times \mathcal{Y})^\star \to \mathbb{R}^{|\mathcal{Y}|}\big\}$ denotes a function class over the extended space. Unlike local ERM approach, (10) learns a common function over extended space and does not require solving an optimization problem for each test instance. That said, since $\mathcal{F}^{\mathrm{ex}}$ operates on the extended feature space, it can be significantly complex and computationally expensive to employ as compared to $\mathcal{F}^{\mathrm{loc}}$.

Our goal is to develop a theoretical understanding of the generalization behavior of these two retrieval-based methods for classification with locally regular data distributions. We present our theoretical treatment of local ERM and classification with extended feature space in Sec. 3 and 4, respectively.

# 3 LOCAL EMPIRICAL RISK MINIMIZATION

Before presenting an excess risk bound for the local ERM method, we introduce various necessary definitions and assumptions that play a critical role in our analysis. We say that a scorer $f$ is $L$-coordinate Lipschitz iff for all $y \in \mathcal{Y}$ and $x_1, x_2 \in \mathcal{X}$, we have

$$|f_y(x) - f_y(x')| \leq L\|x - x'\|_2.$$

In this section, we restrict ourselves to the loss functions that act on the margin of a scorer (cf. (1)), i.e., for any given example $(x, y)$ and any scorer $f$, we have $\ell(f(x), y)) = \ell(\gamma_f(x, y))$. In addition, we assume that, naturally, $\ell$ is a decreasing function of the margin. Furthermore, we assume that $\ell$ is $L_\ell$-Lipschitz function, i.e., $|\ell(\gamma) - \ell(\gamma')| \leq L_\ell |\gamma - \gamma'|, \forall \gamma \geq \gamma'$.

Note that the local ERM selects a scorer from $\mathcal{F}^{\mathrm{loc}}$. At $x \in \mathcal{X}$, let $f^{x,*}$ denote the minimizer of the population version of the local loss, and $f^*$ the population risk minimizer for the global loss, i.e.,

$$f^{x,*} = \arg\min_{f \in \mathcal{F}^{\mathrm{loc}}} \mathbb{E}_{(X',Y') \sim \mathsf{D}^{x,r}}\big[\ell\big(f(X'), Y'\big)\big] \text{ and } f^* = \arg\min_{f \in \mathcal{F}^{\mathrm{global}}} \mathbb{E}_{(X,Y) \sim \mathsf{D}}\big[\ell\big(f(X), Y\big)\big]. \tag{11}$$

Given a distribution $\mathsf{D}$, we define the *weak margin condition* (Döring et al., 2018) for a scorer $f$ as:

**Definition 3.1.** *A scorer $f$ satisfies $(\alpha, c)$-weak margin condition iff, for all $t \geq 0$,*

$$\mathbb{P}_{(X,Y) \sim \mathsf{D}}(|\gamma_f(X, Y)| \leq t) \leq c\, t^\alpha.$$

One of the key assumptions that we rely on is the existence of an underlying scorer $f^{\mathrm{true}}$ that explains the true labels, while ensuring the weak margin condition (cf. Definition 3.1). Here, we note that the true function $f^{\mathrm{true}}$ may neither lie in the function class $\mathcal{F}^{\mathrm{global}}$, nor in $\mathcal{F}^{\mathrm{loc}}$.

**Assumption 3.2** (True scorer function). *There exists a scorer $f^{\mathrm{true}}$ such that for all, $(x, y) \in \mathcal{X} \times \mathcal{Y}$, $f^{\mathrm{true}}$ generates the true label, i.e., $\gamma_{f^{\mathrm{true}}}(x, y) > 0$ and $|\mathcal{R}^X| \perp_{\mathsf{D}} \gamma_{f^{\mathrm{true}}}(X, Y)$. Furthermore, we assume $f^{\mathrm{true}}$ is $L_{\mathrm{true}}$-coordinate Lipschitz, and satisfies the $(\alpha_{\mathrm{true}}, c_{\mathrm{true}})$-weak margin condition.*

### 3.1 Excess risk bound for local ERM

Now that we have introduced the required background and assumptions, we move to presenting our results on characterizing the generalization behavior of local ERM. In particular, we aim to bound

$$\mathbb{E}_{(X,Y)\sim\mathsf{D}}\big[\ell(\hat{f}^X(X),Y) - \ell(f^*(X),Y)\big]. \tag{12}$$

Note that in the above equation $\hat{f}^X$ (cf. (7)) is a function of $\mathcal{R}^X$, and expectation over $\mathcal{R}^X$ is taken implicitly. Towards this, we first obtain the following upper bound on (12).

**Lemma 3.3.** *The expected excess risk of the local ERM optimization $\hat{f}^X$ is bounded as*

$$\mathbb{E}_{(X,Y)\sim\mathsf{D}}\Big[\ell(\hat{f}^X(X),Y) - \ell(f^*(X),Y)\Big] \leq \underbrace{\mathbb{E}_{(X,Y)\sim\mathsf{D}}\Big[\mathbb{E}_{(X',Y')\sim\mathsf{D}^{x,r}}\big[\ell(f^{X,*}(X'),Y') - \ell(f^*(X'),Y')\big]\Big]}_{\textit{Local vs Global Optimal Risk}}$$

$$+ \underbrace{\sum_{\mathcal{F}\in\{\mathcal{F}^{\mathrm{global}},\mathcal{F}^{\mathrm{loc}}\}} \mathbb{E}_{(X,Y)\sim\mathsf{D}}\Big[\sup_{f\in\mathcal{F}}\big|\mathbb{E}_{(X',Y')\sim\mathsf{D}^{x,r}}\big[\ell(f(X'),Y')\big] - \ell(f(X),Y)\big|\Big]}_{\textit{Global and Local: Sample vs Retrieved Set Risk}}$$

$$+ \underbrace{\mathbb{E}_{(X,Y)\sim\mathsf{D}}\Big[\sup_{f\in\mathcal{F}^{\mathrm{loc}}}\Big|\mathbb{E}_{(X',Y')\sim\mathsf{D}^{x,r}}[\ell(f(X'),Y')] - \tfrac{1}{|\mathcal{R}^X|}\sum_{(x',y')\in\mathcal{R}^X}\ell\big(f(x'),y'\big)\Big|\Big]}_{\textit{Generalization of Local ERM}}$$

$$+ \underbrace{\mathbb{E}_{(X,Y)\sim\mathsf{D}}\Big[\Big|\mathbb{E}_{(X',Y')\sim\mathsf{D}^{x,r}}[\ell(f^{X,*}(X'),Y')] - \tfrac{1}{|\mathcal{R}^X|}\sum_{(x',y')\in\mathcal{R}^X}\ell\big(f^{X,*}(x'),y'\big)\Big|\Big]}_{\textit{Central Absolute Moment of } f^{X,*}}.$$

We delegate the proof of Lem. 3.3 to Appendix B. Now, as a strategy to obtain desired excess risk bounds, we separately bound the four terms appearing in Lem. 3.3. Note that the first term captures the expected difference between the loss incurred by global population optima $f^* \in \mathcal{F}^{\mathrm{global}}$ and the local population optima $f^{x,*} \in \mathcal{F}^{\mathrm{loc}}$ in a local region around test instance $x$. The second term aims to capture the loss for a scorer evaluated at $x$ vs. the expected value of the loss for the scorer at a random instance sampled in the local region of $x$ based on $\mathsf{D}^{x,r}$. The third term corresponds to the standard 'generalization error' for the local ERM with respect to the local data distribution $\mathsf{D}^{X,r}$, whereas the fourth term is the empirical variation of the true local function $f^{X,*}$ around its true mean under $\mathsf{D}^{X,r}$.

Let the coordinate-Lipschitz constants for scorers in $\mathcal{F}^{\mathrm{loc}}$ and $\mathcal{F}^{\mathrm{global}}$ be $L_{\mathrm{loc}}$ and $L_{\mathrm{global}}$, respectively. We define a function class $\mathcal{G}(X,Y) = \{(x',y') \mapsto \ell(\gamma_f(\cdot,\cdot)) - \ell(\gamma_f(X,Y)) : f \in \mathcal{F}^{\mathrm{loc}}\}$. Here, by subtracting $\ell\big(f(X),Y\big)$ from the loss, we center the losses on $\mathcal{R}^X$ for any function $f \in \mathcal{F}^{\mathrm{loc}}$, and obtain a tighter bound by utilizing the local nature of the distribution $\mathsf{D}^{X,r}$. For any $L > 0$, for notational convenience let us define

$$\mathcal{M}_r(L;\ell,f_{\mathrm{true}},\mathcal{F}) = 2L_\ell\Big(Lr + \big(\max\{Lr, 2\|\mathcal{F}\|_\infty\} - Lr\big)c_{\mathrm{true}}\big(2L_{\mathrm{true}}r\big)^{\alpha_{\mathrm{true}}}\Big). \tag{13}$$

Now, by controlling different terms appearing in the bound in Lem. 3.3, we obtain the following.

**Theorem 3.4.** *For any $\delta > 0$, the expected excess risk of the local ERM solution $\hat{f}^X$ is bounded as*

$$\mathbb{E}_{(X,Y)\sim\mathsf{D}}\Big[\ell(\hat{f}^X(X),Y) - \ell(f^*(X),Y)\Big]$$

$$\leq \underbrace{(\varepsilon_{\mathcal{X}} + \varepsilon_{\mathrm{loc}})}_{\textit{Local vs Global Optimal loss (I)}} + \underbrace{\mathcal{M}_r(L_{\mathrm{loc}};\ell,f_{\mathrm{true}},\mathcal{F}^{\mathrm{loc}}) + \mathcal{M}_r(L_{\mathrm{global}};\ell,f_{\mathrm{true}},\mathcal{F}^{\mathrm{global}})}_{\textit{Global and Local: Sample vs Retrieved Set Risk (II)}}$$

$$+ \underbrace{2\,\mathbb{E}_{(X,Y)\sim\mathsf{D}}\Big[\mathfrak{R}_{\mathcal{R}^X}\big(\mathcal{G}(X,Y)\big)\Big] + 5\mathcal{M}_r(L_{\mathrm{loc}};\ell,f_{\mathrm{true}},\mathcal{F}^{\mathrm{loc}})\sqrt{\frac{2\ln(4/\delta)}{N(r,\delta)}} + 4\delta L_\ell\|\mathcal{F}^{\mathrm{loc}}\|_\infty(2 + \sqrt{2\ln(4/\delta)})}_{\textit{Generalization of Local ERM (III)}},$$

*where $\mathfrak{R}_{\mathcal{R}^X}\big(\mathcal{G}(X,Y)\big)$ is the empirical Rademacher complexity of $\mathcal{G}(X,Y)$.*

Before discussing the implications of the aforementioned excess risk bound, we instantiate $\mathcal{F}^{\mathrm{loc}}$ with a few common function classes from the literature (see Appendix B for the detailed proof of Thm. 3.4, and about the descriptions of these specific instances).

**Kernel-based classifiers.** When $f_y(\cdot)$ belongs to a bounded RKHS with $\ell_\infty$ norm bound $B$ (Zhang, 2004), for some universal constant $C > 0$ and any $\delta > 0$,

$$\mathbb{E}_{(X,Y)\sim\mathsf{D}}\mathfrak{R}_{\mathcal{R}^X}\big(\mathcal{G}(X,Y)\big) \leq C\big(\sqrt{|\mathcal{Y}|}L_\ell B\ln(n+1)^{3/2}/\sqrt{|N(r,\delta)|} + 2\delta B\big).$$

Similarly, when $f_y(\cdot)$ belongs to a bounded RKHS with $\ell_2$ norm bound $B$ (Lei et al., 2019), for some universal constant $C' > 0$ and any $\delta > 0$,

$$\mathbb{E}_{(X,Y)\sim\mathsf{D}}\mathfrak{R}_{\mathcal{R}^X}\big(\mathcal{G}(X,Y)\big) \leq C'\big(L_\ell B\ln(n|\mathcal{Y}|)^{3/2}/\sqrt{|N(r,\delta)|} + 2\delta B\big).$$

**Feed-forward classifiers.** Assume that $f_y(\cdot)$ is an $L$ layer feed-forward network with 1-Lipschitz non-linearities (Bartlett et al., 2017). Let, for layers $l = 1$ to $L$, the dimension of the weight matrix be $(d_l \times d_{l-1})$ with $d_L = |\mathcal{Y}|$. Also, let $b_l$ and $s_l$ be the $\ell_{2,1}$ norm and spectral norm upper bounds for layer $l$ weight matrix, respectively, with $b_l/s_l \leq \kappa$. We define $d_{\max} = \max_{l\in[L]} d_l$ and let $\tilde{B} = \max_{x\in\mathcal{X}} \|x\|_2 \prod_{l=1}^{L} s_l$. Then, for some universal constant $C'' > 0$ and any $\delta > 0$,

$$\mathbb{E}_{(X,Y)\sim\mathsf{D}}\mathfrak{R}_{\mathcal{R}^X}\big(\mathcal{G}(X,Y)\big) \leq C''\big(L_\ell\tilde{B}\sqrt{\kappa}\ln(d_{\max})L^{3/4}\ln(L_\ell\tilde{B}\sqrt{n})^{3/2}/\sqrt{N(r,\delta)} + 2\delta\tilde{B}\big).$$

**Implications of the excess risk bound.** Our main result for local-ERM highlights the trade-offs in *approximation* vs. *generalization* as the retrieval radius $r$ varies. To further elaborate, note that the approximation error comprises two components, defined by (I) and (II) in Thm. 3.4. $\varepsilon_{\mathcal{X}}$ shows the gap in approximating the $r$-radius neighborhood around $X$ with a simple local function class $\mathcal{F}^X$ which vary with $X \in \mathcal{X}$. $\varepsilon_{\mathrm{loc}}$ shows the gap in approximating the union of the local function class $\cup_{x\in\mathcal{X}}\mathcal{F}^X$ with a single function class $\mathcal{F}^{\mathrm{loc}}$ (possibly with smaller complexity) but while allowing for choosing a different optimizer $f^X \in \mathcal{F}^{\mathrm{loc}}$ for each $X \in \mathcal{X}$. As $r$ increases, both the terms $\varepsilon_{\mathcal{X}}$ and $\varepsilon_{\mathrm{loc}}$ typically increase. For example, in approximating a polynomial function locally with linear function $\varepsilon_{\mathcal{X}}$ increases as the radius increases. Thus, (I) increases with $r$. Note that the second component of the approximation error (II) corresponds to the difference of risk for the sample $X$ and the retrieved set $\mathcal{R}^X$ for $\mathcal{F}^{\mathrm{global}}$ and $\mathcal{F}^{\mathrm{loc}}$, i.e., $\mathcal{M}_r(L_{\mathrm{global}}; \ell, f_{\mathrm{true}}, \mathcal{F}^{\mathrm{global}})$ and $\mathcal{M}_r(L_{\mathrm{loc}}; \ell, f_{\mathrm{true}}, \mathcal{F}^{\mathrm{loc}})$. As we increase $r$, Eq. (13) suggests that the terms increase as $O(\mathrm{poly}(r))$.

On the other hand, the generalization error (III) depends on the size of the retrieved set $\mathcal{R}^X$ and the Rademacher complexity of $\mathcal{G}(X,Y)$ which is induced by $\mathcal{F}^{\mathrm{loc}}$. With increasing radius $r$, the term $N(r,\delta)$ increases. The Rademacher complexity decays with increasing radius, $r$, typically at the rate of $O(1/\sqrt{N(r,\delta)})$. Thus, under the local ERM setting the total approximation error increases with increasing radius $r$, given $\mathcal{F}^{\mathrm{loc}}$ is fixed. On the contrary, the generalization error decreases with increasing radius $r$ for a fixed $\mathcal{F}^{\mathrm{loc}}$. This suggests a trade-off between the approximation and generalization error as we make a design choice about $r$. (We empirically validate this in Figure 2.)

Also, it's worth comparing local-ERM with conventional (non-local) ERM. Under the local-regularity condition assumption (Sec. 2.2), one would utilize a simple $\mathcal{F}^{\mathrm{loc}}$ for local-ERM, which would correspond to the Rademacher complexity term in Theorem 3.4 being small. In contrast, the generalization bound for the traditional (non-local) ERM approach would depend on the Rademacher complexity of a function class $\mathcal{F}^{\mathrm{global}}$ that can achieve a low approximation error on the *entire domain*. Such a function class (even under the regularity assumption) would be much more complex than $\mathcal{F}^{\mathrm{loc}}$, resulting in a large Rademacher complexity. For the right design choice of $r$, and $\mathcal{F}^{\mathrm{loc}}$, the approximation error increase of local-ERM can be offset by large generalization error of $\mathcal{F}^{\mathrm{global}}$. As a consequence, local ERM with simple function class $\mathcal{F}^{\mathrm{loc}}$ can outperform (non-local) ERM with a complex class $\mathcal{F}^{\mathrm{global}}$.

## 3.2 ENDOWING LOCAL ERM WITH GLOBAL REPRESENTATIONS

Note that the local ERM method takes a somewhat myopic view and does not aim to learn a global hypothesis that (partially or entirely) explains the entire data distribution. Such an approach may potentially result in poor performance in those regions of input domains that are not well represented in the training set. Here, we explore a two-stage learning approach as to leverage the global pattern present in the training data in order to address this apparent shortcoming of local ERM.

Given the training data $\mathcal{S}$ and a simple function class $\mathcal{G}^{\mathrm{loc}} : \mathbb{R}^d \to \mathbb{R}^{|\mathcal{Y}|}$, the first stage involves learning a $d$-dimensional feature map $\Phi_\mathcal{S} : \mathcal{X} \to \mathbb{R}^d$ that simultaneously ensures good representation

for the entire data distribution (Radford et al., 2021; Grill et al., 2020; Cer et al., 2018; Reimers & Gurevych, 2019). Subsequently, given a test instance $x$ and its retrieved neighboring points $\mathcal{R}^x = \{(x'_j, y'_j)\} \subseteq \mathcal{S}$, one employs local ERM with the function class:

$$\mathcal{F}_{\Phi_\mathcal{S}} = \{x \mapsto g \circ \Phi_\mathcal{S}(x) : g \in \mathcal{G}^{\mathrm{loc}}\}. \tag{14}$$

At this point, it is tempting to invoke the proof strategy outlined following Lem. 3.3, with $\mathcal{F}^{\mathrm{loc}}$ replaced with $\mathcal{F}_{\Phi_\mathcal{S}}$ to characterize the performance of the aforementioned two-stage method. Note that one can indeed bound the first two terms appearing in Lem. 3.3 for the two-stage method as well. However, bounding the third term that corresponds to generalization gap for local ERM becomes challenging as $\mathcal{F}_{\Phi_\mathcal{S}}$ depends on $\mathcal{S}$ via the global representation $\Phi_\mathcal{S}$ learned in the first stage. Interestingly, Foster et al. (2019) explored a general framework to address such dependence for standard (non retrieval-based) learning. In fact, as an instantiation of their general framework, Foster et al. (2019, Sec. 5.4) considers the ERM in feature space defined by a representation. We employ their techniques to obtain the following result on the generalization gap for local ERM with $\mathcal{F}_{\Phi_\mathcal{S}}$.

**Proposition 3.5.** *Assume that the representation learned duing the first stage is $\Delta$-sensitive, i.e., for $\mathcal{S}$ and $\mathcal{S}'$ that differ in a single example, we have $\|\Phi_S(x) - \Phi_{S'}(x)\| \leq \Delta \ \forall x \in \mathcal{X}$. Furthermore, we assume that each $g \in \mathcal{G}^{\mathrm{loc}}$ (cf. 14) is $L$-Lipschitz, the loss $\ell : \mathbb{R}^{|\mathcal{Y}|} \times |\mathcal{Y}| \to \mathbb{R}$ is $L_{\ell,1}$-Lipschitz w.r.t. $\|\cdot\|_\infty$-norm in the first argument, and $\ell$ is bounded by $M_\ell$. Then, the following holds with probability at least $1 - \delta$.*

$$\sup_{f \in \mathcal{F}_{\Phi_\mathcal{S}}} \left| \mathbb{E}_{(X',Y') \sim \mathsf{D}^{x,r}}[\ell(f(X'), Y')] - \hat{R}^x_\ell(f) \right| \leq \left( M_\ell + 2\Delta L L_{\ell,1} |\mathcal{R}^x| \right) \sqrt{\frac{\log(1/\delta)}{2|\mathcal{R}^x|}} +$$

$$\mathbb{E}_{\mathcal{R}^x \sim \mathsf{D}^{x,r}} \left[ \sup_{f \in \mathcal{F}_{\Phi_\mathcal{S}}} \left| \mathbb{E}_{(X',Y') \sim \mathsf{D}^{x,r}}[\ell(f(X'), Y')] - \hat{R}^x_\ell(f) \right| \right]. \tag{15}$$

*Furthermore* $\quad \mathbb{E}_{\mathcal{R}^x \sim \mathsf{D}^{x,r}} \left[ \sup_{f \in \mathcal{F}_{\Phi_\mathcal{S}}} \left| \mathbb{E}_{(X',Y') \sim \mathsf{D}^{x,r}}[\ell(f(X'), Y')] - \hat{R}^x_\ell(f) \right| \right] \leq 2\mathfrak{R}^\diamond(\ell \circ \mathcal{F}_{\Phi_\mathcal{S}}),$ (16)

*where $\ell \circ \mathcal{F}_{\Phi_\mathcal{S}} = \{(x, y) \mapsto \ell(f(x), y) : f \in \mathcal{F}_{\Phi_\mathcal{S}}\}$ and $\mathfrak{R}^\diamond$ denotes the Rademacher complexity of data dependent hypothesis sets Foster et al. (2019).*

We defer the proof of Prop. 3.5 and necessary background on Foster et al. (2019) to Appendix C.

As a potential advantage of utilizing a global representation with local ERM, one can realize high-performance local learning with an even simpler function class. For example, it's a common approach to only train a linear classifier on learned representations. Furthermore, a high-quality global representation can ensure good performance for those local regions that are not well represented in the training set. We leave a formal treatment of these topics for a longer version of this manuscript.

# 4 CLASSIFICATION IN EXTENDED FEATURE SPACE

Next, we focus on a family of retrieval-based methods that directly learn a scorer to map an input instance and its neighboring labeled instance to a score vector (cf. (10)). In fact, as discussed in Sec. 1, many successful modern instances of retrieval-based models such as REINA (Wang et al., 2022) and KATE (Liu et al., 2022) belong to this family. In this section, we provide the first rigorous treatment (to the best of our knowledge) for such models.

Note that our objective is to learn a function $f : \mathcal{X} \times (\mathcal{X} \times \mathcal{Y})^\star \to \mathbb{R}^{|\mathcal{Y}|}$ (cf. Sec. 2.3). In this work, we restrict ourselves to a sub-family of such retrieval-based methods that first map $\mathcal{R}^x \sim \mathsf{D}^{x,r}$ to $\hat{\mathsf{D}}^{x,r}$ — an empirical estimate of the local distribution $\mathsf{D}^{x,r}$, which is subsequently utilized to make a prediction for $x$. In particular, the scorers of interest are of the form:

$$(x, \mathcal{R}^x) \mapsto f(x, \hat{\mathsf{D}}^{x,r}) = \left( f_1(x, \hat{\mathsf{D}}^{x,r}), \ldots, f_{|\mathcal{Y}|}(x, \hat{\mathsf{D}}^{x,r}) \right) \in \mathbb{R}^{|\mathcal{Y}|}, \tag{17}$$

Note that the general framework for learning in the extended feature space $\widetilde{\mathcal{X}} := \mathcal{X} \times \Delta_{\mathcal{X} \times \mathcal{Y}}$ provides a very rich class of functions. Here, we focus on a specific form of learning methods in $\widetilde{\mathcal{X}}$ by using the kernel methods, adapting the work on kernel methods for domain generalization (Deshmukh et al., 2019). In particular, we study generalization of a kernel-based classifier over $\widetilde{\mathcal{X}}$ learnt via regularized ERM. Due to space constraint, we present an informal version of our result below. See Appendix D for the precise statement (cf. Thm. D.4), necessary background, and detailed proof.

**Theorem 4.1** (Informal). *Let $0 \leq \delta \leq 1$ and $N(r, \delta)$ be as defined in (8). Then, under appropriate assumptions, with probability at least $1 - \delta$, we have*

$$\sup_{f \in \mathcal{F}} \left| \widehat{R}_\ell^{\mathrm{ex}}(f) - R_\ell^{\mathrm{ex}}(f) \right| \lesssim C_1 n^{-\frac{1}{2}} \left( 1 + \log^{\frac{3}{2}} \sqrt{2} n |\mathcal{Y}| \right) + C_2 \sqrt{\frac{\log(\frac{n}{\delta})}{N(r, \frac{\delta}{n})}} + C_3 \sqrt{\frac{\log(\frac{1}{\delta})}{n}},$$

*where $\mathcal{F}$ is the extended feature kernel function class; and $\widehat{R}_\ell^{\mathrm{ex}}(f)$ and $R_\ell^{\mathrm{ex}}(f)$ are empirical and population risks, respectively.*

Interestingly, the bound in Thm. 4.1 implies that the size of the retrieved set $\mathcal{R}^x$ (as captured by $N(r, \frac{\delta}{n})$) has to scale at least logarithmically in the size of the training set $n$ to ensure convergence.

## 5 EXPERIMENTS

There have been numerous successful practical applications of retrieval-based models in the literature (e.g., Wang et al., 2022; Das et al., 2021). Here, we present a brief empirical study for such models in order to corroborate the benefits predicted by our theoretical results.

**Task and dataset.** We perform experiments on both synthetic and real datasets, as summarized below. Further details are relegated to Appendix E.

*(i) Synthetic.* We consider a task of binary classification on a Gaussian mixture. Each mixture component is endowed with its local linear decision boundary. We randomly generate a train set of $n = 10000$ in a 10-dimensional space. We use Euclidean distance for retrieval and perform a 10-fold cross-validation.

*(ii) CIFAR-10.* Next, we consider a task of binary classification on a *real data* for object detection. In particular, we consider a subset of CIFAR-10 dataset where we only restrict to images from "Cat" and "Dog" classes. We randomly partition the data into a train set of $n = 10000$ points and remaining 2000 points for test. We use Euclidean distance for retrieval and do a 10-fold cross-validation.

*(iii) ImageNet.* Finally, we consider 1000-way classification task on ImageNet dataset. We use the standard train-test split with $n = 1281167$ training and 50000 test examples. Following standard practice in literature, we use unsupervised but globally learned features from ALIGN (Jia et al., 2021) to do image retrieval. This also showcases benefits of endowing local ERM with global representation (Sec. 3.2). Given large computational cost, we could only run each experiment once in this setting.

**Methods** On all datasets, as baseline, we consider simple linear classifier and multi-layer perceptron (MLP) of two layers. For retrieval-based models, we consider each of the above methods as the local model to fit on retrieved data points via local ERM framework (Sec. 3). For synthetic datasets, we also considered support vector machines with polynomial kernel (of degree 3) and with radial basis function (RBF) kernel, both for baseline and local ERM. For ImageNet, we additionally consider the state-of-the-art (SoTA) single model published for this task, which is from the most recent CVPR 2022 (Zhai et al., 2022) as a baseline. In addition, for ImageNet, we also consider the pretrain-finetune version of local ERM, where using the retrieved set we fine-tune a MobileNetV3 (Howard et al., 2019) model that has been pretrained on entire ImageNet.

**Observations.** In Fig. 2, we observe the tradeoff of varying the size of the retrieved set (as dictated by the neighborhood radius) on the performance of retrieval-based methods across all settings. We see that when the number of retrieved samples is small, local ERM has lower accuracy, this is due to large generalization error. When the size of the retrieved sample space is high, local ERM fails to minimize the loss effectively due to the lack of model capacity. We see that this effect being more pronounced for simpler function classes such as linear classifier as compared to MLP. In Fig. 2c, we see that, via local ERM with a small MobileNet-V3 model, we are able to achieve the top-1 accuracy of 82.78 whereas a regularly trained MobileNet-V3 model achieves the top-1 accuracy of only 65.80. Also the result is very competitive with SoTA of 90.45 with a *much larger model*. Thus, our empirical evaluation demonstrates the utility of retrieval-based models via simple local ERM framework. In particular, it allows small sized models to attain very high performance.

## 6 RELATED WORK AND DISCUSSION

**Local polynomial regression.** Perhaps the most similar problem to our setup is the rich set of work on local polynomial regression, which has been around for a long time since the pioneering works

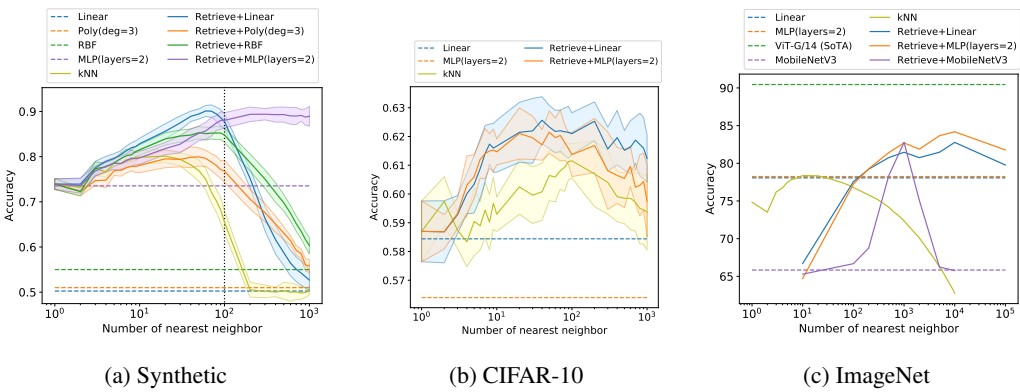

(a) Synthetic      (b) CIFAR-10      (c) ImageNet

Figure 2: Performance of local ERM with size of retrieved set across models of different complexity.

of Stone (1977; 1980). This line of work aims to fit a low-degree polynomial at each point in the data set based on a subset of data points. Such approaches gained a lot of attention as parametric regression was not adequate in various practical applications of the time. The performance of this approach critically depends on subset selected to locally fit the data. Towards this, various selection approaches have been considered: fixed bandwidth (Katkovnik & Kheisin, 1979), nearest neighbors (Cleveland, 1979), kernel weighted (Ruppert & Wand, 1994), and adaptive methods (Ruppert et al., 1995). So far, the analysis of local polynomial regression has been mainly restricted to classical techniques like minimax estimation, on which the literature is a vast for various settings. First results on asymptotic minimax risks were established by Pinsker (1980) over Bobolov spaces. Minimax risks over more general classes were studied by Ibragimov & Has Minskii (2013), Donoho & Liu (1988), among others, for estimating an entire function. But none of these works provide finite sample generalization bounds, which we obtain in this work.

**Multi-task and meta learning** At a surface level, our setup might resemble multi-task and meta learning frameworks. In multi-task learning, we are given the examples from $T$ tasks/distributions and the objective is to ensure good classification performance on all the tasks. In meta-learning, the setting is made harder by requiring good performance on a new target task. As a common approach in these settings, we learn a shared representation across the tasks and then learn a simple task-specific mapping on top of these learned shared features (Vilalta & Drissi, 2002, interalia). While there is a vast literature on multi-task and meta-learning methods, the number of theoretical investigations is quite limited. There are a few works studying upper-bounds on generalization error in multi-task environments (Amit & Meir, 2017; Ben-David & Borbely, 2008; Ben-David et al., 2010; Pentina & Lampert, 2014), and even fewer in case of meta-learning (Balcan et al., 2019; Khodak et al., 2019; Tripuraneni et al., 2021; Du et al., 2020). However, most of these works assume linear or other classes of very simple models, whereas we consider general function class using kernel methods. Moreover, recall that our assumption on the underlying data distribution (Sec. 2.2) implies that it can be approximated by a mixture of tasks. However, by design most of these tasks have a very little overlap in the instance space. Additionally, the number of tasks can be very large in our case. Finally, it's not a priori clear which task a particular example belongs to. Thus, it is not straightforward to employ the aforementioned representation based approach for multi-task or meta-learning approaches for our setting. Interestingly, in this work, we show that retrieval-based approach alleviate the needs to identify the task-membership. By relying on retrieved neighboring instance, it is possible to obtain performance guarantees on their data domain which are attuned to local structure of the problem (cf. Sec. 3).

**Conclusion and future direction.** In this work, we initiate the development of a theoretical framework to study the generalization behavior of retrieval-based modern machine learning models. Our treatment of an explicit local learning paradigm, namely local-ERM, establishes an approximation vs. generalization error trade-off. This highlights the advantage realized by access to a retrieved set during classification as it enables good performance with much simpler (local) function classes. As for the retrieval-based models that leverage a retrieved set without explicitly performing local learning, we present a systematic study by considering a kernel-based classifier over extended feature space. Studying end-to-end retrieval-based models beyond kernel-based classification is a natural and fruitful direction for future work. It's also worth exploring if existing retrieval-based end-to-end models inherently perform *implicit* local learning via architectures such as Transformers.

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

## A PRELIMINARIES

**Definition A.1** (Rademacher complexity). *Given a sample $\mathcal{S} = \{z_i = (x_i, y_i)\}_{i \in [n]} \subset \mathcal{Z}$ and a real-valued function class $\mathcal{F} : \mathcal{Z} \to \mathbb{R}$, the* empirical *Rademacher complexity of $\mathcal{F}$ with respect to $\mathcal{S}$ is defined as*

$$\mathfrak{R}_{\mathcal{S}}(\mathcal{F}) = \frac{1}{n} \mathbb{E}_{\boldsymbol{\sigma}} \left[ \sup_{f \in \mathcal{F}} \sum_{i=1}^{n} \sigma_i f(z_i) \right], \tag{18}$$

*where $\boldsymbol{\sigma} = \{\sigma_i\}_{i \in [n]}$ is a collection of $n$ i.i.d. Bernoulli random variables. For $n \in \mathbb{N}$, the Rademacher complexity $\bar{\mathfrak{R}}_n(\mathcal{F})$ and* worst case *Rademacher complexity $\mathfrak{R}_n(\mathcal{F})$ are defined as follows.*

$$\bar{\mathfrak{R}}_n(\mathcal{F}) = \mathbb{E}_{\mathcal{S} \sim \mathsf{D}^n} [\mathfrak{R}_{\mathcal{S}}(\mathcal{F})], \quad and \quad \mathfrak{R}_n(\mathcal{F}) = \sup_{\mathcal{S} \sim \mathcal{Z}^n} \mathfrak{R}_{\mathcal{S}}(\mathcal{F}). \tag{19}$$

**Definition A.2** (Covering Number). *Let $\epsilon > 0$ and $\| \cdot \|$ be a norm defined over $\mathbb{R}^n$. Given a function class $\mathcal{F} : \mathcal{Z} \to \mathbb{R}$ and a collection of points $\mathcal{S} = \{z_i\}_{i \in [n]} \subset \mathcal{Z}$, we call a set of points $\{u_j\}_{j \in [m]} \subset \mathbb{R}^n$ an $(\epsilon, \| \cdot \|)$-cover of $\mathcal{F}$ with respect to $\mathcal{S}$, if we have*

$$\sup_{f \in \mathcal{F}} \min_{j \in [m]} \|f(\mathcal{S}) - u_j\| \le \epsilon, \tag{20}$$

*where $f(\mathcal{S}) = (f(z_1), \ldots, f(z_n)) \in \mathbb{R}^n$. The $\| \cdot \|$-covering number $\mathcal{N}_{\|\cdot\|}(\epsilon, \mathcal{F}, \mathcal{S})$ denotes the cardinally of the minimal $(\epsilon, \| \cdot \|)$-cover of $\mathcal{F}$ with respect to $\mathcal{S}$. In particular, if $\| \cdot \|$ is an normalized-$\ell_p$ norm ($\|v\| = (\frac{1}{dim(v)} \sum_{i=1}^{dim(v)} |v_i|^p)^{1/p}$), then we simply use $\mathcal{N}_p(\epsilon, \mathcal{F}, \mathcal{S})$ to denote the corresponding $\ell_p$-covering number.*

## B PROOFS FOR SECTION 3.1

### B.1 PROOF OF LEMMA 3.3

Note that

$\mathbb{E}_{(X,Y) \sim \mathsf{D}} \left[ \ell(\hat{f}^X(X), Y) - \ell(f^*(X), Y) \right]$

// We add and subtract loss of the local optimizer $f^{X,*}(\cdot)$ expected over $\mathsf{D}^{X,r}$

$= \mathbb{E}_{(X,Y) \sim \mathsf{D}} \left[ \ell(\hat{f}^X(X), Y) - \mathbb{E}_{(X',Y') \sim \mathsf{D}^{X,r}} \left[ \ell(f^{X,*}(X'), Y') \right] \right.$
$\left. + \mathbb{E}_{(X',Y') \sim \mathsf{D}^{X,r}} \left[ \ell(f^{X,*}(X'), Y') \right] - \ell(f^*(X), Y) \right]$

// We add and subtract loss of the global optimizer $f^*(\cdot)$ expected over $\mathsf{D}^{X,r}$

$= \mathbb{E}_{(X,Y) \sim \mathsf{D}} \left[ \ell(\hat{f}^X(X), Y) - \mathbb{E}_{(X',Y') \sim \mathsf{D}^{X,r}} \left[ \ell(f^{X,*}(X'), Y') \right] \right.$
$+ \mathbb{E}_{(X',Y') \sim \mathsf{D}^{X,r}} \left[ \ell(f^*(X'), Y') \right] - \ell(f^*(X), Y)$
$\left. + \mathbb{E}_{(X',Y') \sim \mathsf{D}^{X,r}} \left[ \ell(f^{X,*}(X'), Y') \right] - \mathbb{E}_{(X',Y') \sim \mathsf{D}^{X,r}} \left[ \ell(f^*(X'), Y') \right] \right]$

// We group (1) local vs global optimizer, (2) global optimizer at $X$ vs expected over $\mathsf{D}^{X,r}$,

// and (3) ERM loss at $X$ vs local optimizer loss expected over $\mathsf{D}^{X,r}$

$= \mathbb{E}_{(X,Y) \sim \mathsf{D}} \left[ \mathbb{E}_{(X',Y') \sim \mathsf{D}^{X,r}} \left[ \ell(f^{X,*}(X'), Y') - \ell(f^*(X'), Y') \right] \right]$
$+ \mathbb{E}_{(X,Y) \sim \mathsf{D}} \left[ \mathbb{E}_{(X',Y') \sim \mathsf{D}^{X,r}} \left[ \ell(f^*(X'), Y') \right] - \ell(f^*(X), Y) \right]$
$+ \mathbb{E}_{(X,Y) \sim \mathsf{D}} \left[ \ell(\hat{f}^X(X), Y) - \mathbb{E}_{(X',Y') \sim \mathsf{D}^{X,r}} \left[ \ell(f^{X,*}(X'), Y') \right] \right]$

// We add and subtract loss of the empirical optimizer $\hat{f}^X(\cdot)$ expected over $\mathsf{D}^{X,r}$

$= \mathbb{E}_{(X,Y) \sim \mathsf{D}} \left[ \mathbb{E}_{(X',Y') \sim \mathsf{D}^{X,r}} \left[ \ell(f^{X,*}(X'), Y') - \ell(f^*(X'), Y') \right] \right]$

$$+ \mathbb{E}_{(X,Y)\sim\mathsf{D}}\Big[\mathbb{E}_{(X',Y')\sim\mathsf{D}^{X,r}}\big[\ell\big(f^*(X'),Y'\big)\big] - \ell\big(f^*(X),Y\big)\Big]$$

$$+ \mathbb{E}_{(X,Y)\sim\mathsf{D}}\Big[\ell\big(\hat{f}^X(X),Y\big) - \mathbb{E}_{(X',Y')\sim\mathsf{D}^{X,r}}[\ell\big(\hat{f}^X(X'),Y'\big)]$$

$$+ \mathbb{E}_{(X',Y')\sim\mathsf{D}^{X,r}}[\ell\big(\hat{f}^X(X'),Y'\big)] - \mathbb{E}_{(X',Y')\sim\mathsf{D}^{X,r}}[\ell\big(f^{X,*}(X'),Y'\big)]\Big]$$

// We (1) bound difference of loss at $X$ and loss expected over $\mathsf{D}^{X,r}$ by maximizing over function class,

// and (2) Subtract empirical loss of empirical optimizer and add (larger) empirical loss of local optimizer

$$\leq \mathbb{E}_{(X,Y)\sim\mathsf{D}}\Big[\mathbb{E}_{(X',Y')\sim\mathsf{D}^{X,r}}\big[\ell\big(f^{X,*}(X'),Y'\big) - \ell\big(f^*(X'),Y'\big)\big]\Big]$$

$$+ \mathbb{E}_{(X,Y)\sim\mathsf{D}}\Big[\sup_{f\in\mathcal{F}^{\mathrm{global}}}\big|\mathbb{E}_{(X',Y')\sim\mathsf{D}^{X,r}}\big[\ell\big(f(X'),Y'\big)\big] - \ell\big(f(X),Y\big)\big|\Big]$$

$$+ \mathbb{E}_{(X,Y)\sim\mathsf{D}}\Big[\sup_{f\in\mathcal{F}^{\mathrm{loc}}}\big|\ell\big(f(X),Y\big) - \mathbb{E}_{(X',Y')\sim\mathsf{D}^{X,r}}[\ell\big(f(X'),Y'\big)]\big|\Big]$$

$$+ \mathbb{E}_{(X,Y)\sim\mathsf{D}}\Big[\mathbb{E}_{(X',Y')\sim\mathsf{D}^{X,r}}[\ell\big(\hat{f}^X(X'),Y'\big)] - \frac{1}{|\mathcal{R}^X|}\sum_{(x',y')\in\mathcal{R}^X}\ell\big(\hat{f}^X(x'),y'\big)\Big]$$

$$+ \mathbb{E}_{(X,Y)\sim\mathsf{D}}\Big[\frac{1}{|\mathcal{R}^X|}\sum_{(x',y')\in\mathcal{R}^X}\ell\big(f^{X,*}(x'),y'\big) - \mathbb{E}_{(X',Y')\sim\mathsf{D}^{X,r}}[\ell\big(f^{X,*}(X'),Y'\big)]\Big] \quad (21)$$

// We (1) bound difference of empirical vs expected loss of empirical optimizer by maximizing over function class,

$$\leq \mathbb{E}_{(X,Y)\sim\mathsf{D}}\Big[\mathbb{E}_{(X',Y')\sim\mathsf{D}^{X,r}}\big[\ell\big(f^{X,*}(X'),Y'\big) - \ell\big(f^*(X'),Y'\big)\big]\Big]$$

$$+ \mathbb{E}_{(X,Y)\sim\mathsf{D}}\Big[\sup_{f\in\mathcal{F}^{\mathrm{global}}}\big|\mathbb{E}_{(X',Y')\sim\mathsf{D}^{X,r}}\big[\ell\big(f(X'),Y'\big)\big] - \ell\big(f(X),Y\big)\big|\Big]$$

$$+ \mathbb{E}_{(X,Y)\sim\mathsf{D}}\Big[\sup_{f\in\mathcal{F}^{\mathrm{loc}}}\big|\ell\big(f(X),Y\big) - \mathbb{E}_{(X',Y')\sim\mathsf{D}^{X,r}}[\ell\big(f(X'),Y'\big)]\big|\Big]$$

$$+ \mathbb{E}_{(X,Y)\sim\mathsf{D}}\Big[\sup_{f\in\mathcal{F}^{\mathrm{loc}}}\big|\mathbb{E}_{(X',Y')\sim\mathsf{D}^{X,r}}[\ell\big(f(X'),Y'\big)] - \frac{1}{|\mathcal{R}^X|}\sum_{(x',y')\in\mathcal{R}^X}\ell\big(f(x'),y'\big)\big|\Big]$$

$$+ \mathbb{E}_{(X,Y)\sim\mathsf{D}}\Big[\big|\mathbb{E}_{(X',Y')\sim\mathsf{D}^{X,r}}\big[\ell\big(f^{X,*}(X'),Y'\big)\big] - \frac{1}{|\mathcal{R}^X|}\sum_{(x',y')\in\mathcal{R}^X}\ell\big(f^{X,*}(x'),y'\big)\big|\Big] \quad (22)$$

$$\square$$

### B.2 PROOF OF THEOREM 3.4

As discussed in Sec. 3, the proof of Theorem 3.4 requires bounding three terms in Lemma 3.3. We now proceed to establishing the desired bounds.

**Local vs global loss.** The local vs global loss can bounded easily using the local regularity condition, and due to the fact that $\mathcal{F}^{\mathrm{loc}} \approx \cup_x \mathcal{F}^x$. Let

$$f^{X,\mathrm{loc}} = \arg\min_{f\in\mathcal{F}^X} \mathbb{E}_{(X',Y')\sim\mathsf{D}^{X,r}}\big[\ell\big(f(X'),Y'\big)\big].$$

$$\mathbb{E}_{(X,Y)\sim\mathsf{D}}\Big[\mathbb{E}_{(X',Y')\sim\mathsf{D}^{X,r}}\big[\ell\big(f^{X,*}(X'),Y'\big) - \ell\big(f^*(X'),Y'\big)\big]\Big]$$

$$\leq \mathbb{E}_{(X,Y)\sim\mathsf{D}}\Big[\mathbb{E}_{(X',Y')\sim\mathsf{D}^{X,r}}\big[\ell\big(f^{X,*}(X'),Y'\big) - \ell\big(f^{X,\mathrm{loc}}(X'),Y'\big)\big]\Big]$$

$$+ \mathbb{E}_{(X,Y)\sim\mathsf{D}}\Big[\mathbb{E}_{(X',Y')\sim\mathsf{D}^{X,r}}\big[\ell\big(f^{X,\mathrm{loc}}(X'),Y'\big) - \ell\big(f^*(X'),Y'\big)\big]\Big]$$

$$\leq \varepsilon_{\mathrm{loc}} + \varepsilon_{\mathcal{X}}.$$

**Global and local: Sample vs retrieved set risk.** The following lemma bounds the second term in Lemma 3.3. Recall the definition, for any $L > 0$,

$$\mathcal{M}_r(L;\ell,f_{\mathrm{true}},\mathcal{F}) = 2L_\ell\Big(Lr + \big(\max\{Lr, 2\|\mathcal{F}\|_\infty\} - Lr\big)c_{\mathrm{true}}\big(2L_{\mathrm{true}}r\big)^{\alpha_{\mathrm{true}}}\Big). \quad (23)$$

**Lemma B.1.** *Under Assumption 3.2, for a L-coordinate Lipschitz function class $\mathcal{F}$ with $\|\mathcal{F}\|_\infty := \sup_{x\in\mathcal{X}}\sup_{f\in\mathcal{F}}\|f(x)\|_\infty$ we have*

$$\mathbb{E}_{(X,Y)\sim\mathsf{D}}\Big[\sup_{f\in\mathcal{F}}\big|\ell(f(X),Y) - \mathbb{E}_{(X',Y')\sim\mathsf{D}^{X,r}}[\ell\big(f(X'),Y'\big)]\big|\Big]$$

$$\leq 2L_\ell\Big(Lr + \big(\max\{Lr, 2\|\mathcal{F}\|_\infty\} - Lr\big)c_{\text{true}}(2L_{\text{true}}r)^{\alpha_{\text{true}}}\Big).$$

*Proof.* We are given the example $(X,Y)$. Let us fix an arbitrary $f \in \mathcal{F}$, and any arbitrary example $(x',y')$ in the $r$ neighborhood of $X$.

We first bound the perturbation in $\gamma_f(\cdot)$ for a given label $\tilde{Y}$.

$$|\gamma_f(X_1,\tilde{Y})) - \gamma_f(X_2,\tilde{Y})| \leq |f_{\tilde{Y}}(X_1) - \max_{s\neq\tilde{Y}} f_s(X_1) - f_{\tilde{Y}}(X_2) + \max_{s'\neq\tilde{Y}} f_{s'}(X_2)|$$

$$\leq |f_{\tilde{Y}}(X_1) - f_{\tilde{Y}}(X_2)| + |\max_{s\neq\tilde{Y}} f_s(X_1) - \max_{s'\neq\tilde{Y}} f_{s'}(X_2)|$$

$$\leq |f_{\tilde{Y}}(X_1) - f_{\tilde{Y}}(X_2)| + \max_{s\neq\tilde{Y}}|f_s(X_1) - f_s(X_2)|$$

$$\leq 2L\|X_1 - X_2\|_2$$

We can now proceed with bounding the loss.

$$|\ell(f(X),Y) - \ell(f(x'),y')| = |\ell(\gamma_f(X,Y)) - \ell(\gamma_f(x',y'))|$$

$$\leq L_\ell|\gamma_f(X,Y) - \gamma_f(x',y')|$$

$$\leq \begin{cases} 4L_\ell\|f\|_\infty; Y\neq y' \\ 2L_\ell Lr; Y = y' \end{cases}$$

Under Assumption 3.2, if we have $\gamma_{f^{\text{true}}}(X,Y) > 2L_{\text{true}}r$, then following the above argument we have $\gamma_{f^{\text{true}}}(X',Y) > 0$, thus $Y$ is the true label of $X'$. In other words, $\gamma_{f^{\text{true}}}(X,Y) > 2L_{\text{true}}r$ imply for any $X'$ in the $r$ neighborhood of $X$ its true label $Y' = Y$.

$$|\ell(f(X),Y) - \ell(f(x'),y')|$$

$$\leq 2L_\ell Lr \mathbb{1}(\gamma_{f^{\text{true}}}(X,Y) > 2L_{\text{true}}r) + 2L_\ell \max\{r, 2\|f\|_\infty\}\mathbb{1}(\gamma_{f^{\text{true}}}(X,Y) \leq 2L_{\text{true}}r)$$

$$\leq 2L_\ell Lr + 2L_\ell\big(\max\{Lr, 2\|f\|_\infty\} - Lr\big)\mathbb{1}(\gamma_{f^{\text{true}}}(X,Y) \leq 2L_{\text{true}}r)$$

As $(x',y')$ was an arbitrary $r$-neighbor, we have

$$|\ell(f(X),Y) - \mathbb{E}_{(X',Y')\sim\mathsf{D}^{X,r}}\ell(f(X'),Y')|$$

$$\leq \mathbb{E}_{(X',Y')\sim\mathsf{D}^{X,r}}|\ell(f(X),Y) - \ell(f(X'),Y')|$$

$$\leq 2L_\ell Lr + 2L_\ell\big(\max\{Lr, 2\|f\|_\infty\} - Lr\big)\mathbb{1}(\gamma_{f^{\text{true}}}(X,Y) \leq 2L_{\text{true}}r)$$

Furthermore, as $f$ was arbitrary, we have

$$\sup_{f\in\mathcal{F}}|\ell(f(X),Y) - \mathbb{E}_{(X',Y')\sim\mathsf{D}^{X,r}}\ell(f(X'),Y')|$$

$$\leq \sup_{f\in\mathcal{F}} 2L_\ell Lr + 2L_\ell\big(\max\{Lr, 2\|f\|_\infty\} - Lr\big)\mathbb{1}(\gamma_{f^{\text{true}}}(X,Y) \leq 2L_{\text{true}}r)$$

$$= 2L_\ell Lr + 2L_\ell\big(\max\{Lr, 2\|\mathcal{F}\|_\infty\} - Lr\big)\mathbb{1}(\gamma_{f^{\text{true}}}(X,Y) \leq 2L_{\text{true}}r).$$

Note $f^{\text{true}}$ is independent of $f$, which was used in the derivation of above inequalities. Taking expectation over $(X,Y)$, and using the margin condition as given in assumption 3.2 we obtain

$$\mathbb{E}_{(X,Y)\sim\mathsf{D}}\Big[\sup_{f\in\mathcal{F}}|\ell(f(X),Y) - \mathbb{E}_{(X',Y')\sim\mathsf{D}^{X,r}}\ell(f(X'),Y')|\Big]$$

$$= 2L_\ell Lr + 2L_\ell\big(\max\{Lr, 2\|\mathcal{F}\|_\infty\} - Lr\big)\mathbb{P}_{(X,Y)\sim\mathsf{D}}\Big[\gamma_{f^{\text{true}}}(X,Y) \leq 2L_{\text{true}}r\Big]$$

$$\leq 2L_\ell Lr + 2L_\ell\big(\max\{Lr, 2\|\mathcal{F}\|_\infty\} - Lr\big)c_{\text{true}}(2L_{\text{true}}r)^{\alpha_{\text{true}}} = \mathcal{M}_r(L;\ell, f_{\text{true}}, \mathcal{F}).$$

$$\square$$

Plugging in the Lipschitz bounds for the function classes $\mathcal{F}^{\text{loc}}$ and $\mathcal{F}^{\text{global}}$ in the above lemma bounds the second term.

**Generalization of Local ERM.** Recall the function class $\mathcal{G}(X,Y) = \{\ell(\gamma_f(\cdot,\cdot)) - \ell(\gamma_f(X,Y)) : f \in \mathcal{F}^{\text{loc}}\}$. Here $\mathcal{G}(X,Y) : \mathcal{X} \times \mathcal{Y} \to \mathbb{R}$. Note that the function class is parameterized by $(X,Y)$. Let us define some quantities of the function class on a set $S \subseteq \mathcal{X} \times \mathcal{Y}$ as

$$\mathcal{G}_{\max}((X,Y); S) = \sup_{g \in \mathcal{G}(X,Y)} \sup_{(x',y') \in S} |g(x',y')|$$

By centering each function $f \in \mathcal{F}^{\text{loc}}$ at the point $(X,Y)$ we can transform the generalization over the function class $\mathcal{F}^{\text{loc}}$, to the generalization over the function class $\mathcal{G}(X,Y)$. In particular, we have

$$\mathbb{E}_{(X,Y)\sim\mathsf{D}}\Big[ \sup_{f\in\mathcal{F}^{\text{loc}}} \Big| \mathbb{E}_{(X',Y')\sim\mathsf{D}^{X,r}}[\ell(f(X'),Y')] - \frac{1}{|\mathcal{R}^X|} \sum_{(x',y')\in\mathcal{R}^X} \ell(f(x'),y') \Big| \Big]$$

$$= \mathbb{E}_{(X,Y)\sim\mathsf{D}}\Big[ \sup_{f\in\mathcal{F}^{\text{loc}}} \Big| \mathbb{E}_{(X',Y')\sim\mathsf{D}^{X,r}}[\ell(f(X'),Y') - \ell(f(X),Y)]$$

$$- \frac{1}{|\mathcal{R}^X|} \sum_{(x',y')\in\mathcal{R}^X} \ell(f(x'),y') - \ell(f(X),Y) \Big| \Big]$$

$$= \mathbb{E}_{(X,Y)\sim\mathsf{D}}\Big[ \sup_{g\in\mathcal{G}(X,Y)} \Big| \mathbb{E}_{(X',Y')\sim\mathsf{D}^{X,r}}[g(X',Y')] - \frac{1}{|\mathcal{R}^X|} \sum_{(x',y')\in\mathcal{R}^X} g(x',y') \Big| \Big].$$

We next state a standard result of learning theory that bounds the final term using the Rademacher complexity of the function class $\mathcal{G}(X,Y)$ (Shalev-Shwartz & Ben-David, 2014).

**Lemma B.2** (Adapted from Theorem 26.5 in Shalev-Shwartz & Ben-David (2014).). *For any* $(X,Y) \in \mathcal{X} \times \mathcal{Y}$ *and a neighborhood set* $\mathcal{R}^X$, *and any function* $g \in \mathcal{G}(X,Y)$, *for each* $\delta > 0$ *with probability at least* $(1 - \delta)$ *the following holds*

$$\Big| \mathbb{E}_{(X',Y')\sim\mathsf{D}^{X,r}}[g(X',Y')] - \frac{1}{|\mathcal{R}^X|} \sum_{(x',y')\in\mathcal{R}^X} g(x',y') \Big|$$

$$\leq 2\mathfrak{R}_{\mathcal{R}^X}(\mathcal{G}(X,Y)) + 4\mathcal{G}_{\max}((X,Y);\mathcal{R}^X)\sqrt{\frac{2\ln(4/\delta)}{|\mathcal{R}^X|}}.$$

Taking expectation with respect to $(X,Y)$, we obtain

$$\mathbb{E}_{(X,Y)\sim\mathsf{D}}\Big[ \sup_{g\in\mathcal{G}(X,Y)} \big| \mathbb{E}_{(X',Y')\sim\mathsf{D}^{X,r}}[g(X',Y')] - \frac{1}{|\mathcal{R}^X|} \sum_{(x',y')\in\mathcal{R}^X} g(x',y') \big| \Big]$$

$$\leq 2\mathbb{E}_{(X,Y)\sim\mathsf{D}}\Big[\mathfrak{R}_{\mathcal{R}^X}(\mathcal{G}(X,Y))\Big] +$$

$$4\mathbb{E}_{(X,Y)\sim\mathsf{D}}\Big[\mathcal{G}_{\max}((X,Y);\mathcal{R}^X)\sqrt{\frac{2\ln(4/\delta)}{|\mathcal{R}^X|}}\Big] + 4\delta L_\ell\|\mathcal{F}^{\text{loc}}\|_\infty$$

$$\leq 2\mathbb{E}_{(X,Y)\sim\mathsf{D}}\Big[\mathfrak{R}_{\mathcal{R}^X}(\mathcal{G}(X,Y))\Big] +$$

$$4\mathbb{E}_{(X,Y)\sim\mathsf{D}}\Big[\mathcal{G}_{\max}((X,Y);\mathcal{R}^X)\Big]\mathbb{E}_{(X,Y)\sim\mathsf{D}}\Big[\sqrt{\frac{2\ln(4/\delta)}{|\mathcal{R}^X|}}\Big] + 4\delta L_\ell\|\mathcal{F}^{\text{loc}}\|_\infty$$

$$\leq 2\mathbb{E}_{(X,Y)\sim\mathsf{D}}\Big[\mathfrak{R}_{\mathcal{R}^X}(\mathcal{G}(X,Y))\Big] + 4\mathcal{M}_r(L_{\text{loc}};\ell,f_{\text{true}},\mathcal{F}^{\text{loc}})\sqrt{\frac{2\ln(4/\delta)}{N(r,\delta)}}$$

$$+ 4\delta L_\ell\|\mathcal{F}^{\text{loc}}\|_\infty\mathbb{E}_{(X,Y)\sim\mathsf{D}}\Big[\sqrt{\frac{2\ln(4/\delta)}{|\mathcal{R}^X|}}\Big|\,|\mathcal{R}^X| \leq N(r,\delta)\Big] + 4\delta L_\ell\|\mathcal{F}^{\text{loc}}\|_\infty$$

$$\leq 2\mathbb{E}_{(X,Y)\sim\mathsf{D}}\Big[\mathfrak{R}_{\mathcal{R}^X}(\mathcal{G}(X,Y))\Big] + 4\mathcal{M}_r(L_{\text{loc}};\ell,f_{\text{true}},\mathcal{F}^{\text{loc}})\sqrt{\frac{2\ln(4/\delta)}{N(r,\delta)}} + 4\delta L_\ell\|\mathcal{F}^{\text{loc}}\|_\infty(1 + \sqrt{2\ln(4/\delta)}).$$

In the first inequality, we condition on retrieved sets of size at least $N(r, \delta)$ which happens with probability at least $\delta$, by assumption. In the second inequality, with probability $(1 - \delta)$ we apply the bound from Lemma B.2, whereas we use the bound $4L_\ell \|\mathcal{F}^{\text{loc}}\|_\infty$ with remaining probability $\delta$. For the second inequality, with probability $\delta$ we use $4L_\ell \|\mathcal{F}^{\text{loc}}\|_\infty$. Further, we use that the $|\mathcal{R}^X| \leq N(r, \delta)$ with probability at least $(1 - \delta)$. Also from the proof of Lemma B.1 we have that

$$\mathcal{G}_{\max}((X, Y); \mathcal{R}^X) \leq 2L_\ell \Big( Lr + \big( \max\{Lr, 2\|\mathcal{F}^{\text{loc}}\|_\infty\} - Lr \big) \mathbb{1}\big(\gamma_{f^{\text{true}}}(X, Y) \leq 2L_{\text{true}}r \big) \Big).$$

Taking expectation with respect to D completes the bound.

**Central Absolute Moment of $f^{X,*}$.** As the function $f^{X,*}$ is *fixed* using centering, and then Hoeffding bound, we can directly bound the remaining term. We have with probability at least $(1 - \delta)$

$$\left| \mathbb{E}_{(X',Y') \sim \mathsf{D}^{X,r}} \big[ \ell\big(f^{X,*}(X'), Y'\big) \big] - \frac{1}{|\mathcal{R}^X|} \sum_{(x',y') \in \mathcal{R}^X} \ell\big(f^{X,*}(x'), y'\big) \right|$$

$$= \left| \mathbb{E}_{(X',Y') \sim \mathsf{D}^{X,r}} \big[ \ell\big(f^{X,*}(X'), Y'\big) - \ell\big(f^{X,*}(X), Y\big) \big] \right.$$

$$\left. - \frac{1}{|\mathcal{R}^X|} \sum_{(x',y') \in \mathcal{R}^X} \ell\big(f^{X,*}(x'), y'\big) - \ell\big(f^{X,*}(X), Y\big) \right|$$

$$\leq \mathcal{G}_{\max}((X, Y); \mathcal{R}^X) \sqrt{\frac{\ln(2/\delta)}{|\mathcal{R}^X|}}$$

Taking expectation similar to the previous case we obtain,

$$\mathbb{E}_{(X,Y) \sim \mathsf{D}} \left[ \left| \mathbb{E}_{(X',Y') \sim \mathsf{D}^{X,r}} \big[ \ell\big(f^{X,*}(X'), Y'\big) \big] - \frac{1}{|\mathcal{R}^X|} \sum_{(x',y') \in \mathcal{R}^X} \ell\big(f^{X,*}(x'), y'\big) \right| \right]$$

$$\leq \mathbb{E}_{(X,Y) \sim \mathsf{D}} \left[ \mathcal{G}_{\max}((X, Y); \mathcal{R}^X) \sqrt{\frac{\ln(2/\delta)}{|\mathcal{R}^X|}} \right]$$

$$\leq \mathcal{M}_r(L_{\text{loc}}; \ell, f_{\text{true}}, \mathcal{F}^{\text{loc}}) \sqrt{\frac{\ln(2/\delta)}{N(r, \delta)}} + 4\delta L_\ell \|\mathcal{F}^{\text{loc}}\|_\infty.$$

This concludes the proof of Theorem 3.4.

## B.3    Bounding the Rademacher Complexity $\mathfrak{R}_{\mathcal{R}^X}\big(\mathcal{G}(X, Y)\big)$

We now derive bounds on the Rademacher complexity of the class $\mathcal{G}(X, Y)$. We use the covering number based bounds for that purpose. We then start by relating it to the covering number of the $\mathcal{F}^{\text{loc}}$ function class. Finally, we provide a bound on the class of functions residing in bounded norm Reproducing Kernel Hilbert Space.

We will use $\mathcal{G}_{\max}(X, Y)$ instead of $\mathcal{G}_{\max}((X, Y); \mathcal{R}^X)$ when the context is clear. Similar to $\mathcal{G}(X, Y)$, we define the function class $\mathcal{G} = \{\ell(\gamma_f(\cdot, \cdot)) : f \in \mathcal{F}^{\text{loc}}\}$ which does not depend on the locality centered around $(X, Y)$. On a set $S \subseteq \mathcal{X} \times \mathcal{Y}$ we can define $\mathcal{G}_{\max}(S) = \sup_{g \in \mathcal{G}} \sup_{(x',y') \in S} |g(x', y')|$.

**Lemma B.3.** *Under Assumption 3.2 we have for any retrieved set within radius $r$ of $X$, $\mathcal{R}^X$, for any $p \geq 1$*

$$\mathfrak{R}_{\mathcal{R}^X}\big(\mathcal{G}(X, Y)\big)$$

$$\leq \inf_{\epsilon \in [0, \mathcal{G}_{p,\max}(X,Y)/2]} \left( 4\epsilon + \frac{12}{\sqrt{|\mathcal{R}^X|}} \int_\epsilon^{\mathcal{G}_{p,\max}(X,Y)/2} \sqrt{\log\big(\tfrac{2\mathcal{G}_{\max}}{\nu}\big) \log\big(\mathcal{N}_p(\nu/2, \mathcal{G}, \mathcal{R}^X)\big)} \, d\nu \right).$$

*Furthermore, we have*

$$\mathfrak{R}_{\mathcal{R}^X}\big(\mathcal{G}(X, Y)\big)$$

$$\leq \inf_{\epsilon \in [0, \mathcal{G}_{\max}(X,Y)/2]} \left( 4\epsilon + \frac{12}{\sqrt{|\mathcal{R}^X|}} \int_\epsilon^{\mathcal{G}_{\max}(X,Y)/2} \sqrt{\log\big(\mathcal{N}_\infty(\nu/2, \mathcal{G}, \mathcal{R}^X \cup \{(X, Y)\})\big)} \, d\nu \right).$$

*Proof.* Given the set $\mathcal{R}^X$, and some function $g \in \mathcal{G}(X,Y)$ let us define for $p \geq 1$

$$\|g\|_{p,\mathcal{R}^X} = \Big( \tfrac{1}{|\mathcal{R}^X|} \sum_{(x',y') \in \mathcal{R}^X} |g(x',y')|^p \Big)^{1/p}.$$

Then, we have $\mathcal{G}_{p,\max}\big((X,Y);\mathcal{R}^X\big) = \max_{g \in \mathcal{G}} \|g\|_{p,\mathcal{R}^X}$ for all $g \in \mathcal{G}(X,Y)$. For the sake of brevity we will use $\mathcal{G}_{p,\max}(X,Y)$ in place of $\mathcal{G}_{p,\max}\big((X,Y);\mathcal{R}^X\big)$. Note that we have from previous definition $\mathcal{G}_{\max}(X,Y) = \mathcal{G}_{\infty,\max}(X,Y) \geq \mathcal{G}_{p,\max}(X,Y)$ for any $p \geq 1$.

Thus using the Chaining method (Shalev-Shwartz & Ben-David, 2014, Chapter 27) we can bound the Radamacher complexity as

$$\mathfrak{R}_{\mathcal{R}^X}\big(\mathcal{G}(X,Y)\big) \leq \inf_{\epsilon \in [0,\mathcal{G}_{p,\max}(X,Y)/2]} \Big( 4\epsilon + \tfrac{12}{\sqrt{|\mathcal{R}^X|}} \int_{\epsilon}^{\mathcal{G}_{p,\max}(X,Y)/2} \sqrt{\log \mathcal{N}_p(\nu, \mathcal{G}(X,Y), \mathcal{R}^X)} d\nu \Big).$$

To finish the proof we need to show, for $p \geq 1$

$$\mathcal{N}_p(\nu, \mathcal{G}(X,Y), \mathcal{R}^X) \leq \mathcal{N}_p(\nu/2, \mathcal{G}, \mathcal{R}^X) \mathcal{N}_p(\nu/2, \mathcal{G}, \{(X,Y)\}).$$

First we fix any $p \geq 1$. Let $\widehat{\mathcal{U}}$ (a set of real numbers) be a $\nu/2$ cover (in $\ell_p$ norm) of $\mathcal{G}$ with respect to $\{(X,Y)\}$. We have $\mathcal{N}_p(\nu, \mathcal{G}(X,Y), \mathcal{R}^X) \leq \frac{2\mathcal{G}_{\max}}{\nu}$ for any $p \geq 1$ and any $\nu > 0$. Further, let $\tilde{\mathcal{U}}$ be a $\nu/2$ cover of $\mathcal{G}$ with respect to $\mathcal{R}^X$. Note for any $\tilde{u} \in \tilde{\mathcal{U}}$ we have $\tilde{u} \in \mathbb{R}^{|\mathcal{R}^X|}$.

Now, we fix any $g' \in \mathcal{G}$. We have at least one $\tilde{u} \in \tilde{\mathcal{U}}$, and $\hat{u} \in \widehat{\mathcal{U}}$ such that

$$\Big( \tfrac{1}{|\mathcal{R}^X|} \sum_{(x',y') \in \mathcal{R}^X} |g'(x',y') - \tilde{u}(x',y')|^p \Big)^{1/p} \leq \nu/2, \text{ and } |g'(X,Y) - \hat{u}| \leq \nu/2.$$

Therefore,

$$\Big( \tfrac{1}{|\mathcal{R}^X|} \sum_{(x',y') \in \mathcal{R}^X} |\big(g'(x',y') - g'(X,Y)\big) - \big(\tilde{u}(x',y') - \hat{u}\big)|^p \Big)^{1/p}$$

$$= \Big( \tfrac{1}{|\mathcal{R}^X|} \sum_{(x',y') \in \mathcal{R}^X} |\big(g'(x',y') - \tilde{u}(x',y')\big) + \big(\hat{u} - g'(X,Y)\big)|^p \Big)^{1/p}$$

$$\leq \Big( \tfrac{1}{|\mathcal{R}^X|} \sum_{(x',y') \in \mathcal{R}^X} |g'(x',y') - \tilde{u}(x',y')|^p \Big)^{1/p} + |\hat{u} - g'(X,Y)|$$

$$\leq \nu/2 + \nu/2 \leq \nu$$

The first inequality follows by applying Minkowski's inequality. Whereas, for the second inequality we apply Jensen's inequality for $(\cdot)^{1/p}$ being a concave function for $p \geq 1$, and applying the appropriate scaling. Therefore, given the covers $\tilde{\mathcal{U}}$ and $\widehat{\mathcal{U}}$, we can construct the set $\mathcal{U}'$ with entries $u' \in \mathbb{R}^{|\mathcal{R}^X|}$ as: $\mathcal{U}' := \{u' = (\tilde{u}(x,y) - \hat{u}) : \tilde{u} \in \tilde{\mathcal{U}}, \hat{u} \in \widehat{\mathcal{U}}\}$. In particular, $|\mathcal{U}'| = |\widehat{\mathcal{U}}||\tilde{\mathcal{U}}|$. As the choice of $g' \in \mathcal{G}$ and $(x',y') \in \mathcal{R}^X$ were arbitrary, we have $\mathcal{U}'$ to be the cover of $\mathcal{G}(X,Y)$.

For $p = \infty$ we can specialize the bound. In particular, consider $\mathcal{U}$ to be a $\nu/2$ cover (in $\ell_\infty$ norm) of $\mathcal{G}$ with respect to $\mathcal{R}^X \cup \{(X,Y)\}$. Then $\mathcal{U}' := \{u' = (\tilde{u}(x,y) - \hat{u}(X,Y)) : \tilde{u} \in \mathcal{U}\}$ creates a (normalized) $\ell_\infty$ cover for $\mathcal{G}$ with respect to $\mathcal{R}^X$. This is true because $\Big( \tfrac{1}{|\mathcal{R}^X|} \sum_{(x',y') \in \mathcal{R}^X} |g'(x',y') - \tilde{u}(x',y')|^p \Big)^{1/p} \leq |g' - \tilde{u}|_\infty = \nu/2$ and $|\hat{u} - g'(X,Y)| \leq |g' - \tilde{u}|_\infty = \nu/2$. This concludes the proof. $\qquad\square$

The first term in the above Lemma is similar to the Chaining based Rademacher bounds (Shalev-Shwartz & Ben-David, 2014, Chapter 28) for $\mathcal{G}$, but the $\epsilon$ (in inf and in the integral) varies in $[0, \mathcal{G}_{\max}(X,Y)]$ instead of $[0, \mathcal{G}_{\max}]$. For small $r$ we have $\mathcal{G}_{\max}(X,Y) << \mathcal{G}_{\max}$, which can be leveraged to give tight bounds in certain situations.

**Example: $\mathcal{F}^{\mathrm{loc}} \equiv \ell_\infty$-bounded RKHS (Zhang, 2004):** Let us consider the setting of Zhang (2004). In this setting, given some Reproducing Kernel Hilbert Space (RKHS) $H$, and a function $\tilde{f} \in H$, we can define the function $\tilde{f}(\cdot) = \tilde{f} \circ h_x$ where for some $h \in H$. We further define the set of functions with bounded norm

$$H_A = \{\tilde{f}(\cdot) \in H : \|\tilde{f}\|_H \sup_{x \in \mathcal{X}} \|h_x\|_H \le A\}.$$

Finally, our local function class can be defined as

$$\mathcal{F}^{\mathrm{loc}} = H_A^{|\mathcal{Y}|} = \{f(\cdot) : f_y(\cdot) \in H_A, \forall y \in \mathcal{Y}\}.$$

We have $\|\mathcal{F}^{\mathrm{loc}}\|_\infty = A$. Recall that loss function for any $y \in \mathcal{Y}$ is given as $\ell(\gamma_f(x, y))$, for any $f \in \mathcal{F}^{\mathrm{loc}}$. We also have for all $y \in \mathcal{Y}$, $|\ell(\gamma_f(x, y)) - \ell(\gamma_{f'}(x, y))| \le 2L_\ell \sup_y |f_y(x) - f'_y(x)|$ (Zhang, 2004, Assumption 15) with $\gamma_A = 2L_\ell$.

Given the above setting, following Lemma 17 in Zhang (2004)[2], we have for a universal constant $c$

$$\log\left(\mathcal{N}_\infty(2L_\ell\nu, \mathcal{G}, \mathcal{R}^X \cup \{(X, Y)\})\right) \le c|\mathcal{Y}|\|\mathcal{F}^{\mathrm{loc}}\|_\infty^2 \frac{\ln(2 + \|\mathcal{F}^{\mathrm{loc}}\|_\infty/\nu) + \ln(|\mathcal{R}^X| + 1)}{\nu^2}.$$

This gives us the following bound for the Rademacher complexity of $\mathcal{F}^{\mathrm{loc}}$

$$\mathfrak{R}_{\mathcal{R}^X} \le O\left(\sqrt{|\mathcal{Y}|} L_\ell \|\mathcal{F}^{\mathrm{loc}}\|_\infty \frac{\ln(|\mathcal{R}^X| + 1)^{3/2}}{\sqrt{|\mathcal{R}^X|}}\right). \tag{24}$$

*Proof of Equation* (24). Without optimizing over $\epsilon$ above, we plug in $\epsilon = \frac{\mathcal{G}_{\max}(X, Y)}{\sqrt{|\mathcal{R}^X|}}$. We obtain

$$\mathfrak{R}_{\mathcal{R}^X}\left(\mathcal{G}(X, Y)\right)$$

$$\le \frac{4\mathcal{G}_{\max}(X, Y)}{\sqrt{|\mathcal{R}^X|}} + \frac{12}{\sqrt{|\mathcal{R}^X|}} \int_{\frac{\mathcal{G}_{\max}(X, Y)}{\sqrt{|\mathcal{R}^X|}}}^{\mathcal{G}_{\max}(X, Y)/2} \sqrt{\log\left(\mathcal{N}_\infty\left(\nu/2, \mathcal{G}, \mathcal{R}^X \cup \{(X, Y)\}\right)\right)} d\nu$$

$$\le \frac{4\mathcal{G}_{\max}(X, Y)}{\sqrt{|\mathcal{R}^X|}} + \frac{48\sqrt{c|\mathcal{Y}|}L_\ell\|\mathcal{F}^{\mathrm{loc}}\|_\infty}{\sqrt{|\mathcal{R}^X|}} \int_{\frac{\mathcal{G}_{\max}(X, Y)}{\sqrt{|\mathcal{R}^X|}}}^{\mathcal{G}_{\max}(X, Y)/2} \sqrt{\frac{\ln(2 + 4L_\ell\|\mathcal{F}^{\mathrm{loc}}\|_\infty/\nu) + \ln(|\mathcal{R}^X| + 1)}{\nu^2}} d\nu$$

$$\le \frac{4\mathcal{G}_{\max}(X, Y)}{\sqrt{|\mathcal{R}^X|}} + \frac{48\sqrt{c|\mathcal{Y}|}L_\ell\|\mathcal{F}^{\mathrm{loc}}\|_\infty}{\sqrt{|\mathcal{R}^X|}} \int_{\frac{\mathcal{G}_{\max}(X, Y)}{\sqrt{|\mathcal{R}^X|}}}^{\mathcal{G}_{\max}(X, Y)/2} \sqrt{\frac{\ln((\mathcal{G}_{\max}(X, Y) + 4L_\ell\|\mathcal{F}^{\mathrm{loc}}\|_\infty)/\nu) + \ln(|\mathcal{R}^X| + 1)}{\nu^2}} d\nu$$

$$\le \frac{4\mathcal{G}_{\max}(X, Y)}{\sqrt{|\mathcal{R}^X|}} + \frac{48\sqrt{c|\mathcal{Y}|}L_\ell\|\mathcal{F}^{\mathrm{loc}}\|_\infty}{\sqrt{|\mathcal{R}^X|}} \int_{\frac{1}{\sqrt{|\mathcal{R}^X|}}}^{1/2} \sqrt{\frac{\ln((1 + 4L_\ell\|\mathcal{F}^{\mathrm{loc}}\|_\infty/\mathcal{G}_{\max}(X, Y))/\nu') + \ln(|\mathcal{R}^X| + 1)}{\nu'^2}} d\nu'$$

$$\le \frac{4\mathcal{G}_{\max}(X, Y)}{\sqrt{|\mathcal{R}^X|}} + \frac{32\sqrt{c|\mathcal{Y}|}L_\ell\|\mathcal{F}^{\mathrm{loc}}\|_\infty}{\sqrt{|\mathcal{R}^X|}} \left(\ln\left((1 + 4L_\ell\|\mathcal{F}^{\mathrm{loc}}\|_\infty/\mathcal{G}_{\max}(X, Y))\sqrt{|\mathcal{R}^X|}\right) + \ln(|\mathcal{R}^X| + 1)\right)^{3/2}$$

We use $\int_x \sqrt{\ln(a/x) + b}/x \, dx = -2/3(\ln(a/x) + b)^{3/2}$ for the final inequality, and ignore the negative part. $\square$

**Example: $\mathcal{F}^{\mathrm{loc}} \equiv \ell_2$ bounded RKHS (Lei et al., 2019):** We consider a fixed kernel $K(x, x') = \langle\phi(x), \phi(x')\rangle$ for $x, x' \in \mathcal{X}$, and let $H_K$ be the RKHS induced by $K$. Let us define the $\ell_{p,q}$ norm for the vectors $W = (w_1, w_2, \ldots, w_{|\mathcal{Y}|}) \in H_K^{|\mathcal{Y}|}$ as $\|(w_1, \ldots, w_{|\mathcal{Y}|})\|_{p,q} = \|(\|w_1\|_p, \ldots, \|w_{|\mathcal{Y}|}\|_p)\|_q$.

For some norm bound $\Lambda > 0$, the local hypothesis space is defined as

$$\mathcal{F}^{\mathrm{loc}} = \{f(\cdot) : f_y(\cdot) = \langle w_y, \phi(\cdot)\rangle, w_y \in H_K, \forall y \in \mathcal{Y}, \|(w_1, \ldots, w_{|\mathcal{Y}|})\|_{2,2} \le \Lambda\}.$$

Recall that we have the loss function class $\mathcal{G} = \{\ell(\gamma_f(\cdot, \cdot)) : f \in \mathcal{F}^{\mathrm{loc}}\}$, where the loss function $\ell(\cdot)$ is assumed to be $L$-Lipschitz continuous w.r.t. $\ell_\infty$ norm.

---

[2]We correct for a typographical error in Zhang (2004), where the $n \equiv |\mathcal{R}^X|$ comes in the denominator of the bound presented in Lemma 17. But Theorem 4 of Zhang (2002) shows this is a typographical error. Indeed, the covering number is not supposed to decrease with increasing number of points.

Given the retrieved set $\mathcal{R}^X$ for some positive integer $n \geq 1$, $\tilde{\mathcal{F}}^X$ after Equation (8) in Lei et al. (2019) induced by $\mathcal{R}^X$. [3] Let the worst case Rademacher complexity of a function class $\mathcal{F}$ over $n$ points be defined as $\mathfrak{R}_n(\mathcal{F})$. Also, for a set $S$ let $\hat{B}(S) = \max_{(x,y)\in S} \sup_{W:\|W\|_{2,2}\leq\Lambda} \langle w_y, \phi(x)\rangle$. We have from Theorem 23 in Lei et al. (2019) that the covering number is bounded as follows: for any set $S = \{(x_i, y_i) : i = 1, \dots, n\}$ of size $n \geq 1$, for any $\varepsilon > 4L\mathfrak{R}_{n|\mathcal{Y}|}(\tilde{\mathcal{F}}^X)$

$$\log\left(\mathcal{N}_\infty(\varepsilon, \mathcal{G}, S)\right) \leq \frac{16n|\mathcal{Y}|L^2(\mathfrak{R}_{n|\mathcal{Y}|}(\tilde{\mathcal{F}}^X))^2}{\varepsilon^2} \log\left(\frac{2en|\mathcal{Y}|\hat{B}(S)L}{\varepsilon}\right).$$

Furthermore, from equation (18) in Lei et al. (2019) we have for any set

$$\frac{\Lambda\max_{(x,y)\in S}\|\phi(x)\|_2}{\sqrt{2n|\mathcal{Y}|}} \leq \mathfrak{R}_{n|\mathcal{Y}|}(\tilde{\mathcal{F}}^X) \leq \frac{\Lambda\max_{(x,y)\in S}\|\phi(x)\|_2}{\sqrt{n|\mathcal{Y}|}}.$$

Therefore, we have for all $\varepsilon \geq 4L\frac{\Lambda\max_{(x,y)\in S}\|\phi(x)\|_2}{\sqrt{2n|\mathcal{Y}|}}$

$$\log\left(\mathcal{N}_\infty(\varepsilon, \mathcal{G}, S)\right) \leq \frac{16\max_{(x,y)\in S}\|\phi(x)\|_2^2\Lambda^2L^2}{\varepsilon^2} \log\left(\frac{2en|\mathcal{Y}|\hat{B}(S)L}{\varepsilon}\right).$$

Plugging this covering number in in our Rademacher bound with $\epsilon \geq 4L\frac{\Lambda\max_{(x,y)\in S}\|\phi(x)\|_2}{\sqrt{2(|\mathcal{R}^X|+1)|\mathcal{Y}|}}$ and taking $S = \mathcal{R}^X \cup \{(X,Y)\}$ we get

$$\mathfrak{R}_{\mathcal{R}^X}\left(\mathcal{G}(X,Y)\right) \leq \inf_{\epsilon\in[0,\mathcal{G}_{\max}(X,Y)/2]} \left(4\epsilon + \frac{12}{\sqrt{|\mathcal{R}^X|}} \int_\epsilon^{\mathcal{G}_{\max}(X,Y)/2} \sqrt{\log\mathcal{N}_\infty(\nu/2, \mathcal{G}, \mathcal{R}^X\cup\{(X,Y)\})}d\nu\right)$$

$$\leq \frac{16\max_{(x,y)\in\mathcal{R}^X\cup\{(X,Y)\}}\|\phi(x)\|_2\Lambda L}{\sqrt{2(|\mathcal{R}^X|+1)|\mathcal{Y}|}} + \frac{12\times16\max_{(x,y)\in\mathcal{R}^X\cup\{(X,Y)\}}\|\phi(x)\|\Lambda L}{\sqrt{|\mathcal{R}^X|}}\times$$

$$\times \int_{\frac{4L\Lambda\max_{(x,y)\in\mathcal{R}^X\cup\{(X,Y)\}}\|\phi(x)\|_2}{\sqrt{2(|\mathcal{R}^X|+1)|\mathcal{Y}|}}}^{\mathcal{G}_{\max}(X,Y)/2} \frac{1}{\nu}\sqrt{\log\left(\frac{4e(|\mathcal{R}^X|+1)|\mathcal{Y}|\hat{B}(\mathcal{R}^X\cup\{(X,Y)\})L}{\nu}\right)}d\nu$$

$$\leq \frac{16\max_{(x,y)\in\mathcal{R}^X\cup\{(X,Y)\}}\|\phi(x)\|_2\Lambda L}{\sqrt{2(|\mathcal{R}^X|+1)|\mathcal{Y}|}} + \frac{8\times16\max_{(x,y)\in\mathcal{R}^X\cup\{(X,Y)\}}\|\phi(x)\|\Lambda L}{\sqrt{|\mathcal{R}^X|}}\times$$

$$\times \left(\log\left(\frac{4\sqrt{2}eL\hat{B}(\mathcal{R}^X\cup\{(X,Y)\})(|\mathcal{R}^X|+1)|\mathcal{Y}|\sqrt{(|\mathcal{R}^X|+1)|\mathcal{Y}|}}{4L\Lambda\max_{(x,y)\in\mathcal{R}^X\cup\{(X,Y)\}}\|\phi(x)\|_2}\right)\right)^{3/2}$$

$$\leq \frac{16\max_{(x,y)\in\mathcal{R}^X\cup\{(X,Y)\}}\|\phi(x)\|_2\Lambda L}{\sqrt{2(|\mathcal{R}^X|+1)|\mathcal{Y}|}} + \frac{8\times16\max_{(x,y)\in\mathcal{R}^X\cup\{(X,Y)\}}\|\phi(x)\|\Lambda L}{\sqrt{|\mathcal{R}^X|}}\times$$

$$\times \left(\log\left(\sqrt{2}e\big((|\mathcal{R}^X|+1)|\mathcal{Y}|\big)^{3/2}\right)\right)^{3/2}$$

In the final inequality we use the fact that

$$\hat{B}(\mathcal{R}^X\cup\{(X,Y)\}) \leq \max_{(x,y)\in\mathcal{R}^X\cup\{(X,Y)\}}\|\phi(x)\|_2 \sup_{W:\|W\|_{2,2}\leq\Lambda}\|W\|_{2,\infty}$$

$$\leq \max_{(x,y)\in\mathcal{R}^X\cup\{(X,Y)\}}\|\phi(x)\|_2\Lambda$$

Therefore, the final bound on the Rademacher complexity can be given as

$$\mathfrak{R}_{\mathcal{R}^X} \leq O\left(L_\ell\|\mathcal{F}^{\text{loc}}\|_\infty \frac{\ln(|\mathcal{Y}||\mathcal{R}^X|)^{3/2}}{\sqrt{|\mathcal{R}^X|}}\right). \tag{25}$$

**Example: $\mathcal{F}^{\text{loc}} \equiv L$-layer Fully Connected Deep Neural Network (DNN)**(Bartlett et al., 2017): Following Bartlett et al. (2017), we consider a $L$-layer deep neural network (DNN) $f_\mathcal{A} = \sigma_L(A^L\sigma_{L-1}(A^{L-1}\sigma_{L-2}(\dots A^1x))$ for $x \in \mathcal{X}$ where $\mathcal{A} = (A_1, A_2, \dots, A_L)$ is the sequence

---

[3] We need $\tilde{\mathcal{F}}^X$ only to state some theorems in Lei et al. (2019). We refer interested readers to Lei et al. (2019) for the details.

of weight matrices. The matrix $A^l \in \mathbb{R}^{d_{l-1} \times d_l}$ for $l = 1$ to $L$, with $d_L = |\mathcal{Y}|$, and $d_0 = d$ given $\mathcal{X} \subseteq \mathbb{R}^d$. Furthermore, $\sigma_l(\cdot) : \mathbb{R}^{d_l} \to \mathbb{R}^{d_l}$ denotes the non-linearity (including pooling and activation), $\sigma_l$-s are taken to be 1-Lipschitz, and $\sigma_l(0) = 0$. We assume that the $A^l$ matrix is initialized at $M^l$, for each $l = 1$ to $L$. We consider the local function class

$$\mathcal{F}^{\mathrm{loc}} = \{f_{\mathcal{A}} : \|A^l - M^l\|_{2,1} \leq b_l, \|A^l\|_\sigma \leq s_l, \ \forall l \leq l \leq L-1\}.$$

Furthermore, we have for any $f \in \mathcal{F}^{\mathrm{loc}}$ and any $x \in \mathcal{X}$ the function $(f(x), y) \to \ell(\gamma_f(\cdot, \cdot))$ is $2L_\ell$-Lipschitz. Therefore, for a fixed set $S$, we have from Theorem 3.3 in Bartlett et al. (2017) that the covering number of the $\mathcal{G} = \{\ell(\gamma_f(\cdot, \cdot)) : f_{\mathcal{A}} \in \mathcal{F}^{\mathrm{loc}}\}$ is given as

$$\log\left(\mathcal{N}_2(\varepsilon, \mathcal{G}, S)\right) \leq \frac{4L_\ell^2 B^2 ln(2d_{\max}^2)}{\varepsilon^2} \big(\prod_{l=1}^L s_l\big)^2 \big(\sum_{l=1}^L (b_l/s_l)^{2/3}\big)^{3/2} = \frac{R}{\varepsilon^2},$$

where $d_{\max} = \max_{l=1}^L d_l$, $\sqrt{\frac{1}{|S|} \sum_{x \in S} \|x\|_2^2} \leq B$, and

$$R = 4L_\ell^2 B^2 ln(2d_{\max}^2) \big(\prod_{l=1}^L s_l\big)^2 \big(\sum_{l=1}^L (b_l/s_l)^{2/3}\big)^{3/2}.$$

Using a the covering number based bound on Rademacher complexity we obtain
$\mathfrak{R}_{\mathcal{R}^X}\big(\mathcal{G}(X,Y)\big)$

$$\leq \inf_{\epsilon \in [0, \mathcal{G}_{2,\max}(X,Y)/2]} \left(4\epsilon + \frac{12}{\sqrt{|\mathcal{R}^X|}} \int_\epsilon^{\mathcal{G}_{2,\max}(X,Y)/2} \sqrt{\log(\frac{4L_\ell B \prod_{l=1}^L s_l}{\nu}) \log\left(\mathcal{N}_2\big(\nu/2, \mathcal{G}, \mathcal{R}^X\big)\right)} d\nu\right)$$

$$\leq \inf_{\epsilon \in [0, \mathcal{G}_{2,\max}(X,Y)/2]} \left(4\epsilon + \frac{12}{\sqrt{|\mathcal{R}^X|}} \int_\epsilon^{\mathcal{G}_{\max}(X,Y)/2} \sqrt{\log(\frac{4L_\ell B \prod_{l=1}^L s_l}{\nu}) \frac{R}{\nu^2}} d\nu\right)$$

$$\leq \inf_{\epsilon \in [0, \mathcal{G}_{2,\max}(X,Y)/2]} \left(4\epsilon + \frac{8\sqrt{R}}{\sqrt{|\mathcal{R}^X|}} \log^{3/2}\big(\frac{4L_\ell B \prod_{l=1}^L s_l}{\epsilon}\big)\right) - \frac{8\sqrt{R}}{\sqrt{|\mathcal{R}^X|}} \log^{3/2}\big(\frac{8L_\ell B \prod_{l=1}^L s_l}{\mathcal{G}_{2,\max}(X,Y)}\big)$$

$$\leq \left(\frac{4\mathcal{G}_{2,\max}(X,Y)}{\sqrt{|\mathcal{R}^X|}} + \frac{8\sqrt{R}}{\sqrt{|\mathcal{R}^X|}} \log^{3/2}\big(\frac{4L_\ell B \prod_{l=1}^L s_l \sqrt{|\mathcal{R}^X|}}{\mathcal{G}_{2,\max}(X,Y)}\big)\right) - \frac{8\sqrt{R}}{\sqrt{|\mathcal{R}^X|}} \log^{3/2}\big(\frac{8L_\ell B \prod_{l=1}^L s_l}{\mathcal{G}_{2,\max}(X,Y)}\big)$$

## C  PROOFS FOR SECTION 3.2

This section focuses on providing a proof of Proposition 3.5. It follows the proof technique of (Foster et al., 2019, Eq. (9)). Before presenting the proof of Proposition 3.5, we need to introduce a slight variation of the Rademacher complexity for data-dependent hypothesis set.

Let $\mathcal{Z} = \mathcal{X} \times \mathcal{Y}$. Let $\mathcal{R} = \{z_j^{\mathcal{R}}\}, \mathcal{T} = \{z_j^{\mathcal{T}}\} \in \mathcal{Z}^m$ be two $m$-sized samples and $\boldsymbol{\sigma} \in \{+1, -1\}^m$ be a vector of independent Rademacher variables. Now define $\mathcal{R}_{\mathcal{T}, \boldsymbol{\sigma}} = \{z_j^{\mathcal{R}_{\mathcal{T}, \boldsymbol{\sigma}}}\} \in \mathcal{Z}^m$ such that

$$z_j^{\mathcal{R}_{\mathcal{T}, \boldsymbol{\sigma}}} = \begin{cases} z_j^{\mathcal{R}}, & \text{if } \sigma_j = 1, \\ z_j^{\mathcal{T}}, & \text{if } \sigma_j = -1, \end{cases} \tag{26}$$

i.e., $\mathcal{R}_{\mathcal{T}, \boldsymbol{\sigma}}$ is obtained by replacing $i$-th element of $\mathcal{R}$ by $i$-th element of $\mathcal{T}$ iff $\sigma_i = -1$. Let $\mathcal{U} \in \mathcal{Z}^{n-m}$ be an $m - n$-sized sample; for $\mathcal{R} \in \mathcal{Z}^m$, $\mathcal{S}_{\mathcal{R}} = \mathcal{U} \cup \mathcal{R} \in \mathcal{Z}^n$. Note that, following this notation, we have $\mathcal{S}_{\mathcal{R}_{\mathcal{T}, \boldsymbol{\sigma}}} = \mathcal{U} \cup \mathcal{R}_{\mathcal{T}, \boldsymbol{\sigma}}$. For $\mathcal{S} \in \mathcal{Z}^n$, let $\mathcal{H}(\mathcal{S})$ be a data dependent function class (hypothesis set), which does not depend on the ordering of the elements in $\mathcal{S}$.

**Definition C.1** (Rademacher complexity for data-dependent function class). *Let $\mathcal{H} = \{\mathcal{H}(\mathcal{S})\}_{\mathcal{S} \in \mathcal{Z}^n}$ be a family of data dependent function classes. Given $\mathcal{R} = \{z_{j \in [m]}^{\mathcal{R}}\}, \mathcal{T} = \{z_{j \in [m]}^{\mathcal{T}}\} \sim \mathsf{D}^m$ and $\mathcal{U} = \{z_{m+i}^{\mathcal{U}}\}_{i \in [n-m]}$, the empirical Rademacher complexity $\mathfrak{R}_{\mathcal{U}, \mathcal{R}, \mathcal{T}}^\diamond(\mathcal{H})$ and Rademacher complexity $\mathfrak{R}_{\mathcal{U}, m}^\diamond(\mathcal{H})$ are defined as follows.*

$$\mathfrak{R}_{\mathcal{U}, \mathcal{R}, \mathcal{T}}^\diamond(\mathcal{H}) = \frac{1}{m} \mathbb{E}_{\boldsymbol{\sigma}} \left[ \sup_{h \in \mathcal{H}(\mathcal{S}_{\mathcal{R}_{\mathcal{T}, \boldsymbol{\sigma}}})} \sum_{i=1}^m \sigma_i h(z_i^{\mathcal{T}}) \right]$$

$$\mathfrak{R}_{\mathcal{U}}^\diamond(\mathcal{H}) = \frac{1}{m} \mathbb{E}_{\mathcal{R}, \mathcal{T} \sim \mathsf{D}^m} \left[ \sup_{h \in \mathcal{H}(\mathcal{S}_{\mathcal{R}_{\mathcal{T}, \boldsymbol{\sigma}}})} \sum_{i=1}^m \sigma_i h(z_i^{\mathcal{T}}) \right] \tag{27}$$

### C.1 PROOF OF PROPOSITION 3.5

We are now ready to establish the proof of Proposition 3.5. As discussed above, we extend the proof technique of (Foster et al., 2019, Eq. (9)) to obtain this result. Our setting differs from that of Foster et al. (2019) as the local ERM objective only depends on the retrieve samples $\mathcal{R}^x$ while the function class of interest $\mathcal{F}_\mathcal{S} = \mathcal{F}_{\Phi_\mathcal{S}}$ in (14) depends on the entire training set $\mathcal{S}$ via representation $\Phi_\mathcal{S}$. We suitably modify the proof techniques of Foster et al. (2019) to handle this difference.

Let $|\mathcal{R}^x| := m$ and $\mathcal{U} = \mathcal{S}\backslash\mathcal{R}^x$. For $\mathcal{R}, \mathcal{T} \in \mathcal{Z}^m$, we define

$$\Xi(\mathcal{R}, \mathcal{T}) = \sup_{f \in \mathcal{F}_{\Phi_{\mathcal{U} \cup \mathcal{R}}}} \left| \underbrace{\mathbb{E}_{(X',Y') \sim \mathsf{D}^{x,r}}[\ell(f(X'), Y')]}_{:=R_\ell(f;\mathsf{D}^{x,r})} - \underbrace{\frac{1}{m} \sum_{(x',y') \in \mathcal{T}} \ell(f(x'), y')}_{:=\widehat{R}_\ell(f;\mathcal{T})} \right|$$

$$= \sup_{f \in \mathcal{F}_{\Phi_{\mathcal{U} \cup \mathcal{R}}}} \left| R_\ell(f; \mathsf{D}^{x,r}) - \widehat{R}_\ell(f; \mathcal{T}) \right|.$$

Note that we are interested in bounding

$$\Xi(\mathcal{R}^x, \mathcal{R}^x) = \sup_{f \in \mathcal{F}_{\Phi_\mathcal{S}}} \left| \underbrace{\mathbb{E}_{(X',Y') \sim \mathsf{D}^{x,r}}[\ell(f(X'), Y')]}_{R_\ell(f;\mathsf{D}^{x,r})} - \underbrace{\frac{1}{m} \sum_{(x',y') \in \mathcal{T}} \ell(f(x'), y')}_{\widehat{R}_\ell(f;\mathcal{R}^x)=\widehat{R}_\ell^x(f)} \right|,$$

where we have used the fact that $\mathcal{U} \cup \mathcal{R}^x = \mathcal{S}$. Towards this, we first establish that $\Xi(\mathcal{R}, \mathcal{R})$ satisfies the $\left(\frac{M_\ell}{m} + 2\Delta LL_{\ell,1}\right)$-bounded difference property, i.e., for $\mathcal{R}, \mathcal{R}' \in \mathcal{Z}^m$ that only differ in one element, we have

$$\Xi(\mathcal{R}, \mathcal{R}) - \Xi(\mathcal{R}', \mathcal{R}') \leq \frac{M_\ell}{m} + 2\Delta LL_{\ell,1}. \tag{28}$$

Note that

$$\Xi(\mathcal{R}, \mathcal{R}) - \Xi(\mathcal{R}', \mathcal{R}') \leq \underbrace{\Xi(\mathcal{R}, \mathcal{R}) - \Xi(\mathcal{R}, \mathcal{R}')}_{\text{I}} + \underbrace{\Xi(\mathcal{R}, \mathcal{R}') - \Xi(\mathcal{R}', \mathcal{R}')}_{\text{II}}. \tag{29}$$

Now, we will separately bound the two terms in the RHS. Let $\breve{z} = (\breve{x}, \breve{y}) \in \mathcal{R}\backslash\mathcal{R}'$ and $\breve{z}' = (\breve{x}', \breve{y}') \in \mathcal{R}'\backslash\mathcal{R}$. Thus, we have the following bound on the first term.

$$\begin{aligned}
\text{I} &= \Xi(\mathcal{R}, \mathcal{R}) - \Xi(\mathcal{R}, \mathcal{R}') \\
&= \sup_{f \in \mathcal{F}_{\Phi_{\mathcal{U} \cup \mathcal{R}}}} \left| R_\ell(f; \mathsf{D}^{x,r}) - \widehat{R}_\ell(f; \mathcal{R}) \right| - \sup_{f \in \mathcal{F}_{\Phi_{\mathcal{U} \cup \mathcal{R}}}} \left| R_\ell(f; \mathsf{D}^{x,r}) - \widehat{R}_\ell(f; \mathcal{R}') \right| \\
&\leq \sup_{f \in \mathcal{F}_{\Phi_{\mathcal{U} \cup \mathcal{R}}}} \left| \left| R_\ell(f; \mathsf{D}^{x,r}) - \widehat{R}_\ell(f; \mathcal{R}) \right| - \left| R_\ell(f; \mathsf{D}^{x,r}) - \widehat{R}_\ell(f; \mathcal{R}') \right| \right| \\
&\leq \sup_{f \in \mathcal{F}_{\Phi_{\mathcal{U} \cup \mathcal{R}}}} \left[ R_\ell(f; \mathsf{D}^{x,r}) - \widehat{R}_\ell(f; \mathcal{R}) - R_\ell(f; \mathsf{D}^{x,r}) + \widehat{R}_\ell(f; \mathcal{R}') \right] \\
&= \sup_{f \in \mathcal{F}_{\Phi_{\mathcal{U} \cup \mathcal{R}}}} \left| \widehat{R}_\ell(f; \mathcal{R}') - \widehat{R}_\ell(f; \mathcal{R}) \right| \\
&= \sup_{f \in \mathcal{F}_{\Phi_{\mathcal{U} \cup \mathcal{R}}}} \frac{1}{m} \left| \ell(f(\breve{x}'), \breve{y}') - \ell(f(\breve{x}), \breve{y}) \right| \leq \frac{M_\ell}{m}, \tag{30}
\end{aligned}$$

where the last inequality follows from our boundedness assumption for the loss function $\ell$.

Now we move to term II. Towards this, note that, it follows from the definition of supremum that, for any $\epsilon > 0$, there exists $\tilde{f} \in \mathcal{F}_{\Phi_{\mathcal{U} \cup \mathcal{R}}}$ such that

$$\sup_{f \in \mathcal{F}_{\Phi_{\mathcal{U} \cup \mathcal{R}}}} \left| R_\ell(f; \mathsf{D}^{x,r}) - \widehat{R}_\ell(f; \mathcal{R}') \right| - \epsilon \leq \left| R_\ell(\tilde{f}; \mathsf{D}^{x,r}) - \widehat{R}_\ell(\tilde{f}; \mathcal{R}') \right| \tag{31}$$

Let $\tilde{f} = \tilde{g} \circ \Phi_{\mathcal{U} \cup \mathcal{R}} \in \mathcal{F}_{\Phi_{\mathcal{U} \cup \mathcal{R}}}$ and $\tilde{f}' = \tilde{g} \circ \Phi_{\mathcal{U} \cup \mathcal{R}'} \in \mathcal{F}_{\Phi_{\mathcal{U} \cup \mathcal{R}'}}$. Note that, for any $(x, y) \in \mathcal{Z}$,

$$\left| \ell(\tilde{f}(x), y) - \ell(\tilde{f}'(x), y) \right| = \left| \ell(\tilde{g} \circ \Phi_{\mathcal{U} \cup \mathcal{R}}(x), y) - \ell(\tilde{g} \circ \Phi_{\mathcal{U} \cup \mathcal{R}'}(x), y) \right|$$

$$\overset{(i)}{\leq} L_{\ell,1} \| \tilde{g} \circ \Phi_{\mathcal{U} \cup \mathcal{R}}(x) - \tilde{g} \circ \Phi_{\mathcal{U} \cup \mathcal{R}'}(x) \|_\infty$$

$$\leq L_{\ell,1} \| \tilde{g} \circ \Phi_{\mathcal{U} \cup \mathcal{R}}(x) - \tilde{g} \circ \Phi_{\mathcal{U} \cup \mathcal{R}'}(x) \|_2$$

$$\overset{(ii)}{\leq} L_{\ell,1} L \| \Phi_{\mathcal{U} \cup \mathcal{R}}(x) - \Phi_{\mathcal{U} \cup \mathcal{R}'}(x) \|_2$$

$$\overset{(iii)}{\leq} L_{\ell,1} L \Delta, \tag{32}$$

where we use $L_{\ell,1}$-Lipschitzness of $\ell$ w.r.t. $\| \cdot \|_\infty$ norm, $L$-Lipschitzness of $g$, and $\Delta$-sensitivity of the representation $\Phi$ in $(i)$, $(ii)$, and $(iii)$, respectively.

Now, we have

$$\begin{aligned}
\text{II} &= \Xi(\mathcal{R}, \mathcal{R}') - \Xi(\mathcal{R}', \mathcal{R}') \\
&= \sup_{f \in \mathcal{F}_{\Phi_{\mathcal{U} \cup \mathcal{R}}}} \left| R_\ell(f; \mathsf{D}^{x,r}) - \widehat{R}_\ell(f; \mathcal{R}') \right| - \sup_{f \in \mathcal{F}_{\Phi_{\mathcal{U} \cup \mathcal{R}'}}} \left| R_\ell(f; \mathsf{D}^{x,r}) - \widehat{R}_\ell(f; \mathcal{R}') \right| \\
&\overset{(i)}{\leq} \left| R_\ell(\tilde{f}; \mathsf{D}^{x,r}) - \widehat{R}_\ell(\tilde{f}; \mathcal{R}') \right| + \epsilon - \sup_{f \in \mathcal{F}_{\Phi_{\mathcal{U} \cup \mathcal{R}'}}} \left| R_\ell(f; \mathsf{D}^{x,r}) - \widehat{R}_\ell(f; \mathcal{R}') \right| \\
&\leq \left| R_\ell(\tilde{f}; \mathsf{D}^{x,r}) - \widehat{R}_\ell(\tilde{f}; \mathcal{R}') \right| + \epsilon - \left| R_\ell(\tilde{f}'; \mathsf{D}^{x,r}) - \widehat{R}_\ell(\tilde{f}'; \mathcal{R}') \right| \\
&= \left| \left[ R_\ell(\tilde{f}; \mathsf{D}^{x,r}) - R_\ell(\tilde{f}'; \mathsf{D}^{x,r}) \right] - \left[ \widehat{R}_\ell(\tilde{f}; \mathcal{R}') - \widehat{R}_\ell(\tilde{f}'; \mathcal{R}') \right] \right| + \epsilon \\
&\leq \left| R_\ell(\tilde{f}; \mathsf{D}^{x,r}) - R_\ell(\tilde{f}'; \mathsf{D}^{x,r}) \right| + \left| \widehat{R}_\ell(\tilde{f}; \mathcal{R}') - \widehat{R}_\ell(\tilde{f}'; \mathcal{R}') \right| + \epsilon \\
&\overset{(ii)}{\leq} 2 L_{\ell,1} L \Delta + \epsilon, \tag{33}
\end{aligned}$$

where $(i)$ and $(ii)$ follow from (31) and (32), respectively. Now, since $\epsilon$ in (31) can be chosen arbitrarily small, it follows from (29), (30), and (33) that

$$\Xi(\mathcal{R}, \mathcal{R}) - \Xi(\mathcal{R}', \mathcal{R}') \leq \frac{M_\ell}{m} + 2\Delta L L_{\ell,1},$$

i.e., $\Xi(\mathcal{R}, \mathcal{R})$ indeed satisfies the $\left( \frac{M_\ell}{m} + 2\Delta L L_{\ell,1} \right)$-bounded difference property. Now, it follows from the McDiarmid's inequality that, for $\delta > 0$, we have with probability at least $1 - \delta$:

$$\Xi(\mathcal{R}^x, \mathcal{R}^x) \leq \mathbb{E} \left[ \Xi(\mathcal{R}^x, \mathcal{R}^x) \right] + \left( M_\ell + 2\Delta L L_{\ell,1} m \right) \sqrt{\frac{\log(1/\delta)}{2m}}$$

or

$$\sup_{f \in \mathcal{F}_{\Phi_S}} \left| R_\ell(f; \mathsf{D}^{x,r}) - \widehat{R}_\ell^x(f) \right| \leq \mathbb{E}_{\mathcal{R}^x} \left| \sup_{f \in \mathcal{F}_{\Phi_S}} \left[ R_\ell(f; \mathsf{D}^{x,r}) - \widehat{R}_\ell^x(f) \right] \right| +$$

$$\left( M_\ell + 2\Delta L L_{\ell,1} m \right) \sqrt{\frac{\log(1/\delta)}{2m}}. \tag{34}$$

Now, first statement of Proposition 3.5 follows from (34) and the fact that $m = |\mathcal{R}^x|$.

It follows from the proof steps in (Foster et al., 2019, Section E.1) that

$$\mathbb{E}_{\mathcal{R}^x} \left[ \sup_{f \in \mathcal{F}_{\Phi_{S = \mathcal{U} \cup \mathcal{R}^x}}} \left| R_\ell(f; \mathsf{D}^{x,r}) - \widehat{R}_\ell^x(f) \right| \right] \leq 2 \mathfrak{R}_{\mathcal{U}}^\diamond (\ell \circ \mathcal{F}), \tag{35}$$

where $\mathcal{F} = \{ \mathcal{F}_{\Phi_{\mathcal{U} \cup \mathcal{R}}} \}_{\mathcal{R} \in \mathcal{Z}^m}$ and $\mathfrak{R}_{\mathcal{U}}^\diamond$ is defined in (27). This completes the proof of Proposition 3.5. $\square$

## D  CLASSIFICATION IN EXTENDED FEATURE SPACE: A KERNEL-BASED APPROACH

As introduced in Sec. 2.3, our objective is to learn a function $f : \mathcal{X} \times (\mathcal{X} \times \mathcal{Y})^\star \to \mathbb{R}^{|\mathcal{Y}|}$. For a given instance $x$, such a function can leverage its neighboring set $\mathcal{R}^x \in (\mathcal{X} \times \mathcal{Y})^\star$ to improve the prediction on $x$. In this work, we restrict ourselves to a sub-family of such retrieval-based methods that first map

$\mathcal{R}^x \sim \mathsf{D}^{x,r}$ to $\hat{\mathsf{D}}^{x,r}$ — an empirical estimate of the local distribution $\mathsf{D}^{x,r}$, which is subsequently utilized to make a prediction for $x$. In particular, the scorers of interest are of the form:

$$(x, \mathcal{R}^x) \mapsto f(x, \hat{\mathsf{D}}^{x,r}) = \big(f_1(x, \hat{\mathsf{D}}^{x,r}), \ldots, f_{|\mathcal{Y}|}(x, \hat{\mathsf{D}}^{x,r})\big) \in \mathbb{R}^{|\mathcal{Y}|}, \qquad (36)$$

where $f_y(x, \hat{\mathsf{D}}^{x,r})$ denotes the score assigned to the $y$-th class. Thus, assuming that $\Delta_{\mathcal{X} \times \mathcal{Y}}$ denotes the set of distribution over $\mathcal{X} \times \mathcal{Y}$, we restrict to a suitable function class in $\{f : \mathcal{X} \times \Delta_{\mathcal{X} \times \mathcal{Y}} \to \mathbb{R}^{|\mathcal{Y}|}\}$. Note that, given a surrogate loss $\ell : \mathbb{R}^{|\mathcal{Y}|} \times \mathcal{Y} \to \mathbb{R}$ and scorer $f$, the empirical risk $\hat{R}_\ell^{\mathrm{ex}}(f)$ and population risk $R_\ell^{\mathrm{ex}}(f)$ take the following form:

$$\hat{R}_\ell^{\mathrm{ex}}(f) = \frac{1}{n} \sum_{i \in [n]} \ell\big(x_i, \hat{\mathsf{D}}^{x_i,r}\big) \quad \text{and} \quad R_\ell^{\mathrm{ex}}(f) = \mathbb{E}_{(X,Y) \sim \mathsf{D}} \left[ \ell\big(f(X, \mathsf{D}^{X,r}), Y\big) \right]. \qquad (37)$$

Note that that the general framework for learning in the extended feature space $\widetilde{\mathcal{X}} := \mathcal{X} \times \Delta_{\mathcal{X} \times \mathcal{Y}}$ provides a very rich class of functions. In this paper, we focus on a specific form of learning methods in the extended feature space by using the kernel methods. The method as well as its analysis is obtained by adapting the work on utilizing kernel methods for domain generalization (Blanchard et al., 2011; Deshmukh et al., 2019).

## D.1 KERNEL-BASED CLASSIFICATION

Before introducing a kernel method for the classification, we need to define a suitable kernel $k : \widetilde{\mathcal{X}} \times \widetilde{\mathcal{X}} \to \mathbb{R}$ on the extended feature space $\widetilde{\mathcal{X}} := \mathcal{X} \times \Delta_{\mathcal{X} \times \mathcal{Y}}$. Towards this, let $k_{\mathcal{Z}}$ be a kernel over $\mathcal{Z} := \mathcal{X} \times \mathcal{Y}$. Assuming that $H_{k_{\mathcal{Z}}}$ is the reproducing kernel Hilbert space (RKHS) associated with $k_{\mathcal{Z}}$, we can define a kernel mean embedding (Smola et al., 2007) $\Psi : \Delta_{\mathcal{Z}} \to H_{k_{\mathcal{Z}}}$ as follows:

$$\Psi(P) = \int_{\mathcal{Z}} k_{\mathcal{Z}}(z, \cdot)\, dP. \qquad (38)$$

For an empirical distribution $\hat{\mathsf{D}}^{x,r}$ defined by $\mathcal{R}^x$, kernel embedding in (38) takes the following form.

$$\Psi(\hat{\mathsf{D}}^{x,r}) = \frac{1}{|\mathcal{R}^x|} \sum_{(x',y') \in \mathcal{R}^x} k_{\mathcal{Z}}\big((x', y'), \cdot\big). \qquad (39)$$

Now, using a kernel $k_{\mathcal{X}}$ over $\mathcal{X}$ and a kernel-like function $\kappa$ over $\Psi(\Delta_{\mathcal{Z}})$, we define a desired kernel $k : \widetilde{\mathcal{X}} \times \widetilde{\mathcal{X}} \to \mathbb{R}$ as follows:

$$k\big(\widetilde{X}_1, \widetilde{X}_2\big) = k\big((X_1, \mathsf{D}^{X_1,r}), (X_2, \mathsf{D}^{X_2,r})\big) = k_{\mathcal{X}}(X_1, X_2) \cdot \kappa\big(\Psi(\mathsf{D}^{X_1,r}), \Psi(\mathsf{D}^{X_2,r})\big). \qquad (40)$$

Let $H_k$ be the RKHS corresponding to the kernel $k$ in (40), and $\|\cdot\|_{H_k}$ be the norm associated with $H_k$. Equipped with the kernel in (40) and associated $H_k$, for $\lambda > 0$, we propose to learn a scorer $f = (f_1, \ldots, f_{|\mathcal{Y}|}) \in H_k^{|\mathcal{Y}|} := H_k \times \cdots \times H_k$ via the following regularized ERM problem.

$$\hat{f}^{\mathrm{ex}} = \arg \min_{f \in H_k^{|\mathcal{Y}|}} \frac{1}{n} \sum_{i=1}^{n} \ell\big(f(\tilde{x}_i), y_i\big) + \lambda \cdot \Omega(f), \qquad (41)$$

where $\tilde{x}_i = (x_i, \hat{\mathsf{D}}^{x_i,r})$ and $\Omega(f) := \|f\|_{H_k^{|\mathcal{Y}|}}^2 := \sum_{y \in \mathcal{Y}} \|f_y\|_{H_k}^2$. It follows from the representer theorem that the solution of (41) takes the form $\hat{f}^{\mathrm{ex}}(\cdot) = \sum_{i \in [n]} \alpha_i k\big((x_i, \hat{\mathsf{D}}^{x_i,r}), \cdot\big)$. One can apply multiclass extensions of SVMs to learn the weights $\{\alpha_i\}$ (Deshmukh et al., 2019). Next, we focus on studying the generalization behavior of the scorer $\hat{f}^{\mathrm{ex}}$ recovered in (41).

## D.2 GENERALIZATION BOUNDS FOR KERNEL-BASED CLASSIFICATION

Before presenting a generalization bound for kernel-based classification over the extended feature space $\widetilde{\mathcal{X}}$, we state the three key assumptions that are utilized in our analysis.

**Assumption D.1.** *The loss function* $\ell : \mathbb{R}^{|\mathcal{Y}|} \times \mathcal{Y}$ *is* $L_{\ell,1}$*-Lipschitz w.r.t. the first argument, i.e.,*

$$|\ell(s_1, y) - \ell(s_2, y)| \le L_{\ell,1} \cdot \|s_1 - s_2\|_\infty \qquad \forall s_1, s_2 \in \mathbb{R}^{|\mathcal{Y}|} \text{ and } y \in \mathcal{Y}. \qquad (42)$$

*Furthermore, assume that* $\sup_{(x,y)} \ell(x, y) := M_\ell \le \infty$.

**Assumption D.2.** *Kernels $k_{\mathcal{X}}, k_{\mathcal{Z}}$, and $\kappa$ are bounded by $M_{k_{\mathcal{X}}}, M_{k_{\mathcal{Z}}}$, and $M_\kappa$, respectively.*

**Assumption D.3.** *Let $H_{k_{\mathcal{Z}}}$ and $H_\kappa$ be the RKHS associated with $k_{\mathcal{Z}}$ and $\kappa$, respectively. Then, the canonical feature map $\varphi_\kappa : H_{k_{\mathcal{Z}}} \to H_\kappa$ is $\alpha$-Hölder continuous with $\alpha \in (0, 1]$, i.e.,*

$$\|\varphi_\kappa(h_1) - \varphi_\kappa(h_2)\|_{H_\kappa} \leq L' \cdot \|h_1 - h_2\|_{H_{k_{\mathcal{Z}}}}^\alpha \quad \forall h_1, h_2 \in \{h \in H_{k_{\mathcal{Z}}} \ : \ \|h\|_{H_{k_{\mathcal{Z}}}} \leq M_{k_{\mathcal{Z}}}\} \quad (43)$$

The following result states our generalization bound for the kernel-based classification method described in Sec. D.1.

**Theorem D.4.** *Let $0 \leq \delta \leq 1$ and Assumptions D.1–D.3 hold. Furthermore, let $N(r, \delta)$ be as defined in (8). Then, for any $B > 0$, the following holds with probability at least $1 - 3\delta$*

$$\sup_{f \in \mathcal{F}_B^k} \left| \widehat{R}_\ell^{\mathrm{ex}}(f) - R_\ell^{\mathrm{ex}}(f) \right| \leq 32\sqrt{\log 2} L_{\ell,1} B M_\kappa M_{k_{\mathcal{X}}} n^{-\frac{1}{2}} \left( 1 + \log^{\frac{3}{2}} \sqrt{2} n |\mathcal{Y}| \right)$$

$$+ L_{\ell,1} L' M_{k_{\mathcal{X}}} B \left( M_{k_{\mathcal{Z}}} \sqrt{\frac{2\log(\frac{n}{\delta})}{N(r, \frac{\delta}{n})}} + M_{k_{\mathcal{Z}}} \sqrt{\frac{1}{N(r, \frac{\delta}{n})}} + \frac{4 M_{k_{\mathcal{Z}}} \log(\frac{n}{\delta})}{3 N(r, \frac{\delta}{n})} \right)^\alpha + M\sqrt{\frac{\log(\frac{1}{\delta})}{2n}},$$

*where $\mathcal{F}_B^k = \left\{ f = (f_1, \ldots, f_{|\mathcal{Y}|}) \in H_k^{|\mathcal{Y}|} : \Omega(f) \leq B^2 \right\}$ and $M := M_\ell + L_{\ell,1} B M_{k_{\mathcal{X}}} M_\kappa$.*

Before presenting the proof of Theorem D.4, we state two key results from the literature that are used in our analysis.

**Proposition D.5** (Steinwart & Christmann (2008)). *Let $(\Omega, \mathcal{A}, P)$ be a probability space, $H$ be a separable Hilbert space, and $M > 0$. Let $\eta_1, \ldots, \eta_m : \Omega \to H$ be $m$ independent $H$-valued random variables satisfying $\|\eta_j\|_\infty \leq M$, for all $j \in [m]$. The, for $\delta > 0$, the following holds with probability at least $1 - \delta$.*

$$\left\| \frac{1}{m} \sum_{j=1}^m (\eta_j - \mathbb{E}_P[\eta_j]) \right\|_H \leq M\sqrt{\frac{2\log(1/\delta)}{m}} + M\sqrt{\frac{1}{m}} + \frac{4M\log(1/\delta)}{3m}. \quad (44)$$

**Proposition D.6.** *(Deshmukh et al., 2019; Lei et al., 2019) Let $\widetilde{\mathcal{Z}} = \widetilde{\mathcal{X}} \times \mathcal{Y}$ be (extended) input and output space pair and $\widetilde{\mathcal{S}} = \left\{ \tilde{z}_1, \ldots, \tilde{z}_n \right\}$. Let $H_k$ be a RKHS defined on $\widetilde{\mathcal{X}}$, with $k$ being the associated kernel. Let*

$$\mathcal{F}_B^k = \left\{ (f_1, \ldots, f_{|\mathcal{Y}|}) \ : \ f_y \in H_k \ \forall y \in \mathcal{Y} \text{ and } \Big( \sum_{y \in \mathcal{Y}} \|f_y\|_{H_k}^p \Big)^{1/p} \leq B \right\}$$

*and $\ell : \mathbb{R}^{|\mathcal{Y}|} \times \mathcal{Y} \to \mathbb{R}$ be a Lipschitz function in its first argument, i.e.,*

$$|\ell(s_1, y) - \ell(s_2, y)| \leq L_{\ell,1} \|s_1 - s_2\|_\infty \quad \forall s_1, s_2 \in \mathbb{R}^{|\mathcal{Y}|} \text{ and } y \in \mathcal{Y}.$$

*Then the Rademacher complexity of the induced function class $\ell \circ \mathcal{F}_B^k := \{\ell \circ f : f \in \mathcal{F}_B^k\}$ satisfies*

$$\mathfrak{R}_{\widetilde{\mathcal{S}}}\left( \ell \circ \mathcal{F}_B^k \right) := \mathbb{E}_{\sigma_i} \Big[ \sup_{f \in \mathcal{F}_B^k} \frac{1}{n} \sum_{i \in [n]} \sigma_i \ell\big(f(\tilde{x}_i), y_i\big) \Big]$$

$$\leq 16 L_{\ell,1} \sqrt{\log 2} B \sup_{\tilde{x} \in \widetilde{\mathcal{X}}} \sqrt{k(\tilde{x}, \tilde{x})} n^{-\frac{1}{2}} |\mathcal{Y}|^{\frac{1}{2} - \frac{1}{\max\{2, p\}}} \left( 1 + \log^{\frac{3}{2}} \sqrt{2} n |\mathcal{Y}| \right). \quad (45)$$

*Note that $\boldsymbol{\sigma} = (\sigma_1, \ldots, \sigma_n)$ denotes $n$ i.i.d. Rademacher random variable.*

*Proof of Theorem D.4.* Note that

$$\sup_{f \in \mathcal{F}_B^k} \left| \widehat{R}_\ell^{\mathrm{ex}}(f) - R_\ell^{\mathrm{ex}}(f) \right| = \sup_{f \in \mathcal{F}_B^k} \left| \frac{1}{n} \sum_{i=1}^n \ell\big(f(x_i, \widehat{\mathsf{D}}^{x_i, r}), y_i\big) - \mathbb{E}_{(X,Y) \sim \mathsf{D}} \left[ \ell\big(f(X, \mathsf{D}^{X, r}), Y\big) \right] \right|$$

$$\leq \underbrace{\sup_{f \in \mathcal{F}_B^k} \left| \frac{1}{n} \sum_{i=1}^n \ell\big(f(x_i, \widehat{\mathsf{D}}^{x_i, r}), y_i\big) - \frac{1}{n} \sum_{i=1}^n \ell\big(f(x_i, \mathsf{D}^{x_i, r}), y_i\big) \right|}_{\mathrm{I}} +$$

$$\underbrace{\sup_{f \in \mathcal{F}_B^k} \left| \frac{1}{n} \sum_{i=1}^n \ell\big(f(x_i, \mathsf{D}^{x_i, r}), y_i\big) - \mathbb{E}_{(X,Y) \sim \mathsf{D}} \left[ \ell\big(f(X, \mathsf{D}^{X,r}), Y\big)\right]\right|}_{\text{II}} \quad (46)$$

**Bounding the term-I in** (46). Note that

$$\begin{aligned}
\mathrm{I} &= \sup_{f \in \mathcal{F}_B^k} \left| \frac{1}{n} \sum_{i=1}^n \ell\big(f(x_i, \widehat{\mathsf{D}}^{x_i, r}), y_i\big) - \frac{1}{n} \sum_{i=1}^n \ell\big(f(x_i, \mathsf{D}^{x_i, r}), y_i\big)\right| \\
&\leq \frac{L_{\ell,1}}{n} \sum_{i \in [n]} \|f(x_i, \widehat{\mathsf{D}}^{x_i, r}) - f(x_i, \mathsf{D}^{x_i, r})\|_\infty \\
&\leq \frac{L_{\ell,1}}{n} \sum_{i \in [n]} \max_{y \in \mathcal{Y}} |f_y(x_i, \widehat{\mathsf{D}}^{x_i, r}) - f_y(x_i, \mathsf{D}^{x_i, r})| \\
&\leq L_{\ell,1} \cdot \max_{y \in \mathcal{Y}} \max_{i \in [n]} |f_y(x_i, \widehat{\mathsf{D}}^{x_i, r}) - f_y(x_i, \mathsf{D}^{x_i, r})| \quad (47)
\end{aligned}$$

It follows from the reproducing property of the kernel $k$ that, for any $y \in \mathcal{Y}$,

$$\begin{aligned}
|f_y(x_i, \widehat{\mathsf{D}}^{x_i, r}) - f_y(x_i, \mathsf{D}^{x_i, r})| &= |\langle f_y, k((x_i, \widehat{\mathsf{D}}^{x_i, r}), \cdot) - k((x_i, \mathsf{D}^{x_i, r}), \cdot)\rangle| \\
&\leq \|f_y\|_{H_k} \cdot \|k((x_i, \widehat{\mathsf{D}}^{x_i, r}), \cdot) - k((x_i, \mathsf{D}^{x_i, r}), \cdot)\|_{H_k}. \quad (48)
\end{aligned}$$

Now,

$$\begin{aligned}
&\|k((x_i, \widehat{\mathsf{D}}^{x_i, r}), \cdot) - k((x_i, \mathsf{D}^{x_i, r}), \cdot)\|_{H_k} \\
&= \Big( k((x_i, \widehat{\mathsf{D}}^{x_i, r}), (x_i, \widehat{\mathsf{D}}^{x_i, r})) + k((x_i, \mathsf{D}^{x_i, r})), (x_i, \mathsf{D}^{x_i, r})) - 2k((x_i, \widehat{\mathsf{D}}^{x_i, r}), (x_i, \mathsf{D}^{x_i, r}))\|_{H_k} \Big)^{1/2} \\
&= \sqrt{k_{\mathcal{X}}(x_i, x_i)} \Big( \kappa(\Psi(\widehat{\mathsf{D}}^{x_i, r}), \Psi(\widehat{\mathsf{D}}^{x_i, r})) + \kappa(\Psi(\mathsf{D}^{x_i, r})), \Psi(\mathsf{D}^{x_i, r})) - \\
&\hspace{4cm} 2\kappa(\Psi(\widehat{\mathsf{D}}^{x_i, r}), \Psi(\mathsf{D}^{x_i, r}))\|_{H_k} \Big)^{1/2} \\
&= \sqrt{k_{\mathcal{X}}(x_i, x_i)} \|\kappa(\Psi(\widehat{\mathsf{D}}^{x_i, r}), \cdot) - \kappa(\Psi(\mathsf{D}^{x_i, r}), \cdot)\|_{H_\kappa} \\
&\leq M_{k_{\mathcal{X}}} \|\kappa(\Psi(\widehat{\mathsf{D}}^{x_i, r}), \cdot) - \kappa(\Psi(\mathsf{D}^{x_i, r}), \cdot)\|_{H_\kappa} \quad (49) \\
&= M_{k_{\mathcal{X}}} \|\varphi_\kappa(\Psi(\widehat{\mathsf{D}}^{x_i, r})) - \varphi_\kappa(\Psi(\mathsf{D}^{x_i, r}))\|_{H_\kappa} \\
&\leq L' M_{k_{\mathcal{X}}} \cdot \|\Psi(\widehat{\mathsf{D}}^{x_i, r}) - \Psi(\mathsf{D}^{x_i, r})\|_{H_{k_{\mathcal{Z}}}}^\alpha \quad (50)
\end{aligned}$$

By combining (48) and (49), we obtain that

$$|f_y(x_i, \widehat{\mathsf{D}}^{x_i, r}) - f_y(x_i, \mathsf{D}^{x_i, r})| \leq L' M_{k_{\mathcal{X}}} \cdot \|f_y\|_{H_k} \cdot \|\Psi(\widehat{\mathsf{D}}^{x_i, r}) - \Psi(\mathsf{D}^{x_i, r})\|_{H_{k_{\mathcal{Z}}}}^\alpha. \quad (51)$$

Now, Hoeffding's inequality in Hilbert spaces (cf. Proposition D.5) implies that, for $i \in [n]$, the following holds with probability at least $1 - \delta$.

$$\begin{aligned}
\|\Psi(\widehat{\mathsf{D}}^{x_i, r}) - \Psi(\mathsf{D}^{x_i, r})\|_{H_{k_{\mathcal{Z}}}}^\alpha &= \left\| \frac{1}{|\mathcal{R}^{x_i}|} \sum_{(x', y') \in \mathcal{R}^{x_i}} k_{\mathcal{Z}}((x', y'), \cdot) - \mathbb{E}_{\mathsf{D}^{x_i, r}}\left[k_{\mathcal{Z}}((X', Y'), \cdot)\right] \right\|_{H_{k_{\mathcal{Z}}}} \\
&\leq M_{k_{\mathcal{Z}}} \sqrt{\frac{2\log(1/\delta)}{|\mathcal{R}^{x_i}|}} + M_{k_{\mathcal{Z}}} \sqrt{\frac{1}{|\mathcal{R}^{x_i}|}} + \frac{4 M_{k_{\mathcal{Z}}} \log(1/\delta)}{3|\mathcal{R}^{x_i}|}. \quad (52)
\end{aligned}$$

It follows from (51) and (52) that, for each $i \in [n]$,

$$\begin{aligned}
&|f_y(x_i, \widehat{\mathsf{D}}^{x_i, r}) - f_y(x_i, \mathsf{D}^{x_i, r})| \\
&\leq L' M_{k_{\mathcal{X}}} \cdot \|f_y\|_{H_k} \cdot \left( M_{k_{\mathcal{Z}}} \sqrt{\frac{2\log(\frac{1}{\delta})}{|\mathcal{R}^{x_i}|}} + M_{k_{\mathcal{Z}}} \sqrt{\frac{1}{|\mathcal{R}^{x_i}|}} + \frac{4 M_{k_{\mathcal{Z}}} \log(\frac{1}{\delta})}{3|\mathcal{R}^{x_i}|} \right)^\alpha \quad \forall\, y \in \mathcal{Y} \quad (53)
\end{aligned}$$

holds with probability at least $1 - \delta$. Next, taking union bound over $i \in [n]$ implies that the following holds for all $i \in [n]$ and $y \in \mathcal{Y}$ with probability at least $1 - \delta$.

$$|f_y(x_i, \widehat{\mathsf{D}}^{x_i, r}) - f_y(x_i, \mathsf{D}^{x_i, r})|$$

$$\leq L' M_{k_{\mathcal{X}}} \|f_y\|_{H_k} \left( M_{k_{\mathcal{Z}}} \sqrt{\frac{2\log(n/\delta)}{|\mathcal{R}^{x_i}|}} + M_{k_{\mathcal{Z}}} \sqrt{\frac{1}{|\mathcal{R}^{x_i}|}} + \frac{4 M_{k_{\mathcal{Z}}} \log(n/\delta)}{3|\mathcal{R}^{x_i}|} \right)^{\alpha}. \tag{54}$$

Recall that, for each $i \in [n]$, we have $|\mathcal{R}^{x_i}| \geq N(r, \delta)$ with probability at least $1 - \delta$ (cf. (8)). Using union bound, we have $|\mathcal{R}^{x_i}| \geq N(r, \delta/n)$, $\forall\, i \in [n]$, with probability at least $1 - \delta$. Thus, the following holds for all $i \in [n]$ and $y \in \mathcal{Y}$ with probability at least $1 - 2\delta$

$$|f_y(x_i, \widehat{\mathsf{D}}^{x_i, r}) - f_y(x_i, \mathsf{D}^{x_i, r})|$$
$$\leq L' M_{k_{\mathcal{X}}} \|f_y\|_{H_k} \left( M_{k_{\mathcal{Z}}} \sqrt{\frac{2\log(n/\delta)}{N(r, \delta/n)}} + M_{k_{\mathcal{Z}}} \sqrt{\frac{1}{N(r, \delta/n)}} + \frac{4 M_{k_{\mathcal{Z}}} \log(n/\delta)}{3 N(r, \delta/n)} \right)^{\alpha}. \tag{55}$$

By using $\|f_y\|_{H_k} \leq B$ and combining (47) with (55), we obtain that

$$\mathrm{I} \leq L_{\ell,1} L' M_{k_{\mathcal{X}}} B \left( M_{k_{\mathcal{Z}}} \sqrt{\frac{2\log(n/\delta)}{N(r, \delta/n)}} + M_{k_{\mathcal{Z}}} \sqrt{\frac{1}{N(r, \delta/n)}} + \frac{4 M_{k_{\mathcal{Z}}} \log(n/\delta)}{3 N(r, \delta/n)} \right)^{\alpha} \tag{56}$$

holds with probability at least $1 - 2\delta$.

**Bounding the term-II in** (46). Note that

$$\mathrm{II} = \sup_{f \in \mathcal{F}_B^k} \left| \frac{1}{n} \sum_{i=1}^{n} \ell\big(f(x_i, \mathsf{D}^{x_i, r}), y_i\big) - \mathbb{E}_{(X,Y) \sim \mathsf{D}} \left[ \ell\big(f(X, \mathsf{D}^{X,r}), Y\big) \right] \right| \tag{57}$$

Using the Assumptions D.1 and D.2 and the fact that $f \in \mathcal{F}_B^k$, we can argue that

$$\ell\big(f(x, \mathsf{D}^{x,r}), y\big) = \ell(0, y) + |\ell\big(f(x, \mathsf{D}^{x,r}), y\big) - \ell(0, y)|$$
$$\leq M_\ell + L_{\ell,1} \|f(x, \mathsf{D}^{x,r})\|_\infty$$
$$\leq M_\ell + L_{\ell,1} \max_{y' \in \mathcal{Y}} \big| \langle f_{y'}, k\big((x, \mathsf{D}^{x,r}), \cdot\big) \rangle \big|$$
$$\leq M_\ell + L_{\ell,1} \max_{y' \in \mathcal{Y}} \|f_{y'}\|_{H_k} M_k$$
$$\leq M_\ell + L_{\ell,1} R M_k \leq M_\ell + L_{\ell,1} R M_{k_{\mathcal{X}}} M_\kappa := M$$

Now, it follows from the Azuma-McDiarmid's inequality that the following holds with probability at least $1 - \delta$.

$$\sup_{f \in \mathcal{F}_B^k} \left| \frac{1}{n} \sum_{i=1}^{n} \ell\big(f(x_i, \mathsf{D}^{x_i, r}), y_i\big) - \mathbb{E}_{(X,Y) \sim \mathsf{D}} \left[ \ell\big(f(X, \mathsf{D}^{X,r}), Y\big) \right] \right|$$
$$\leq \mathbb{E} \left[ \sup_{f \in \mathcal{F}_B^k} \left| \frac{1}{n} \sum_{i=1}^{n} \ell\big(f(x_i, \mathsf{D}^{x_i, r}), y_i\big) - \mathbb{E}_{(X,Y) \sim \mathsf{D}} \left[ \ell\big(f(X, \mathsf{D}^{X,r}) Y\big) \right] \right| \right]$$
$$+ M \sqrt{\frac{\log(1/\delta)}{2n}}, \tag{58}$$

Using the standard symmetrization procedure, we get that

$$\mathbb{E} \left[ \sup_{f \in \mathcal{F}_B^k} \left| \frac{1}{n} \sum_{i=1}^{n} \ell\big(f(x_i, \mathsf{D}^{x_i, r}), y_i\big) - \mathbb{E}_{(X,Y) \sim \mathsf{D}} \left[ \ell\big(f(X, \mathsf{D}^{X,r}) Y\big) \right] \right| \right]$$
$$\leq \frac{2}{n} \cdot \mathbb{E}_{(X_i, Y_i) \sim \mathsf{D}} \mathbb{E}_{\sigma_i} \left[ \sum_{i \in [n]} \sigma_i \ell\big(f(x_i, \mathsf{D}^{x_i, r}), y_i\big) \right],$$
$$= 2 \bar{\mathfrak{R}}_{\widetilde{\mathsf{S}}}\big(\ell \circ \mathcal{F}_B^k\big)$$

where $\boldsymbol{\sigma} = (\sigma_1, \ldots, \sigma_n)$ denotes $n$ i.i.d. Rademacher random variables and $\bar{\mathfrak{R}}_{\widetilde{\mathsf{S}}}\big(\ell \circ \mathcal{F}_B^k\big)$ denote the Rademarcher complexity of the function class

$$\ell \circ \mathcal{F}_B^k = \left\{ (x, y, \mathsf{D}^{x,r}) \mapsto \ell\big(f(x, \mathsf{D}^{x,r}), y\big) \;:\; f \in \mathcal{F}_B^k \right\}.$$

Now, using Proposition D.6 with $p = 2$ and Assumption D.2, we have

$$\bar{\mathfrak{R}}_{\tilde{S}}\left(\ell \circ \mathcal{F}_B^k\right) \leq 16 L_{\ell,1}\sqrt{\log 2}B \sup_{\tilde{x}\in\tilde{\mathcal{X}}} \sqrt{k(\tilde{x},\tilde{x})}n^{-\frac{1}{2}}\left(1+\log^{\frac{3}{2}}\sqrt{2}n|\mathcal{Y}|\right)$$

$$\leq 16 L_{\ell,1}\sqrt{\log 2}B M_\kappa M_{k_\mathcal{X}} n^{-\frac{1}{2}}\left(1+\log^{\frac{3}{2}}\sqrt{2}n|\mathcal{Y}|\right) \tag{59}$$

Now, by combining (57), (58), and (59), we obtain that with probability at least $1-\delta$

$$\text{II} \leq 32\sqrt{\log 2}L_{\ell,1}BM_\kappa M_{k_\mathcal{X}}n^{-\frac{1}{2}}\left(1+\log^{\frac{3}{2}}\sqrt{2}n|\mathcal{Y}|\right) + M\sqrt{\frac{\log(1/\delta)}{2n}}. \tag{60}$$

Finally, combining (46), (56) and (60) completes the proof. $\qquad\square$

## E ADDITIONAL DETAILS FOR EXPERIMENTS

### E.1 SYNTHETIC

**Task and data.** We consider the task of binary classification on mixtures using *synthetic data*: In particular, we assume $k = 100$ clusters in a $D = 10$-dimensional space. Each cluster is specified by a mean parameter $\mu_i \in \mathbb{R}^D \sim \mathsf{Uniform}(-10, 10)$ and a classification weight vector $w_i \in \mathbb{R}^d \sim \mathcal{N}(0, \mathbb{I})$ for $i = 1, 2, \cdots, k$. We randomly generate a train set of $n = 10000$ points as follows: To generate a labeled example $(x_j, y_j), j \in [n]$: 1) select a cluster $i$ uniformly at random, and 2) sample $x_j \sim \mathcal{N}(\mu_i, \mathbb{I})$ and its label $y_j = \mathrm{sign}(w_i^T(x_j - \mu_i))$. Additionally, we also generate another set of points as test set using the same procedure.

**Methods** As baseline, we consider models of various complexity, starting from simple linear classifier, to support vector machines with polynomial kernel (of degree 3) and with radial basis function (RBF) kernel, to a multi-layer perceptron (MLP) of two layers. For retrieval-based models, we consider each of the above method as the local model to fit on retrieved data points via local ERM framework (Sec. 3). Additionally, we also report simple kNN baseline. We compare all these methods using classification accuracy on the held out test set. We repeat all the experiments 10 times.

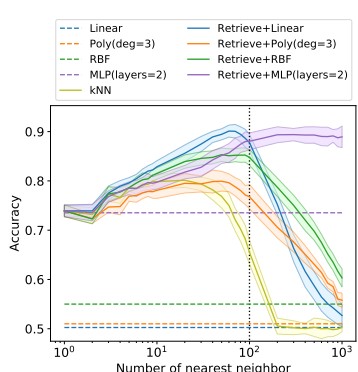

**Observations** In Figure 3, we observe the tradeoff of varying the size of the retrieved set (as dictated by the neighborhood radius) on the performance of the proposed algorithms. We see that when the number of retrieved samples is small the local methods have lower accuracy, this is due to large generalization error. When the size of the retrieved sample space is high, the local methods fail to minimize the loss effectively due to the lack of model capacity. We see that this effect being more pronounced for simpler function classes such as linear classifier as compared to RBF or polynomial classifiers.

Figure 3: Performance of ERM and local ERM for various models on synthetic data.

### E.2 CIFAR-10

**Task and data.** We consider the task of binary classification on a *real image data* for object detection. In particular, we consider a subset of CIFAR-10 dataset where we only restrict to images from "Cat" and "Dog" classes. We randomly partition the data into a train set of $n = 10000$ points and remaining 2000 points for test. We do a 10-fold cross-validation.

**Methods** We consider a subset of method from Appendix. E.1. In particular, we only consider a simple linear classifier and a multi-layer perceptron (MLP) of two layers. For retrieval-based models, we consider each of the above methods as the local model to fit on retrieved data points via local ERM framework (Sec. 3). The retrieval is done using L2 distance in the input space directly (no features is extracted). Additionally, we also report simple kNN baseline. We compare all these methods using classification accuracy on the held out test set. We repeat all the experiments 10 times.

**Observations** Similar to Figure 3, Figure 4 exhibits a tradeoff, where varying the size of the retrieved set (as dictated by the neighborhood radius) impacts the performance of the proposed algorithms. We see when the number of retrieved samples is small the local methods have lower accuracy, this is due to large generalization error; and when the number of retrieved samples is large, simple local function class incurs a large approximation error.

### E.3 IMAGENET

**Task and data.** We consider the task of 1000-way image classification on ImageNet ILSVRC-12 dataset. We use the standard train-test set split, where we have of $n = 1281167$ points for training and $50000$ points for test. Given large computational cost, we could only run each experiment once.

**Methods** We compare proposed Local ERM (Sec. 3) to state-of-the-art (SoTA) single model published for this task, which is from the most recent CVPR 2022 (Zhai et al., 2022). For the local parametric model we use a small MobileNetV3 architecture (Howard et al., 2019) with 4.01M parameters and 156 MFLOPs compute cost. Contrast this to SoTA model ViT-G/14 with 1.84B parameters and 938 GFLOPs compute cost. Following standard practice in literature, we use unsupervised learned features from ALIGN (Jia et al., 2021) to do image retrieval using L2 distance. For solving the local ERM, we fine-tune a MobileNetV3 model, which has been pretrained on ImageNet, on the retrieved set using Adam optimizer with a linear decay schedule. Additionally, we also report simple kNN baseline. We compare all these methods using classification accuracy on the held out test set.

**Observations** In Figure 5, we see that local ERM with a small

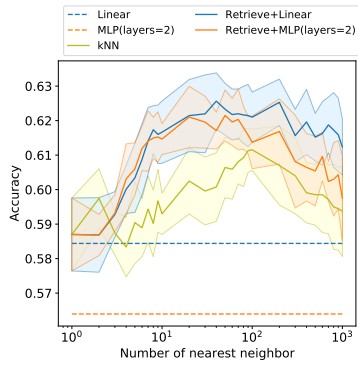

Figure 4: Performance of ERM and local ERM for various models on (binary) CIFAR-10.

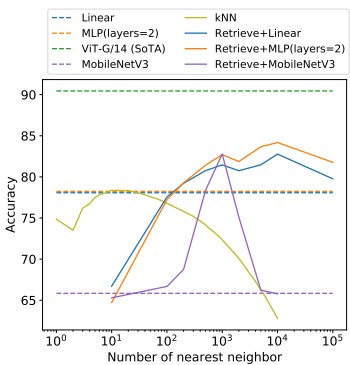

Figure 5: Performance of ERM and local ERM for various models on on ImageNet.

MobileNet-V3 model is able to achieve the top-1 accuracy of 82.78 whereas a regularly trained MobileNet-V3 model achieves the top-1 accuracy of only 65.80. Also the result is very competitive with SoTA of 90.45 with a *much larger model*. Thus, the result suggest that the simple local ERM framework (analyzed in our work) is able to demonstrate the utility of retrieval-based models. In particular, it allows a realistic small sized model to attain very competitive numbers on the popular ImageNet benchmark. Furthermore, as pointed at end of Sec. 3.2, using global representation from ALIGN embeddings help simplest linear model to outperform MobileNet-V3 working directly on image input, thereby showcasing the benefits of endowing local ERM with global representation.

