# OpenReview forum: "Generalization Properties of Retrieval-based Models"
_ICLR.cc/2023/Conference — Submitted to ICLR 2023_

### Official Review · Reviewer_94k1 · 2022-10-23

**Confidence:** 3
**Correctness:** 3
**Technical Novelty And Significance:** 3
**Empirical Novelty And Significance:** Not applicable
**Recommendation:** 6

**Clarity, Quality, Novelty And Reproducibility:**

The presented theoretical and empirical results for generalization properties of retrieval-based models are interesting. This reviewer has to apologize as there may be misunderstanding in this review due to the short review time period but the authors may need to check thoroughly their proofs to make sure that the presentation is correct and self-explaining as much as possible.

It will be more interesting to have the derived results to guide further development of retrieval-based models.

**Strength And Weaknesses:**

Theoretical analysis of retrieval-based models on their generalizability is indeed interesting and timely. The presented empirical experiments have also shown the agreement with the analyses.

It may be interesting to consider how the analysis results may help guide the design and hyperparameter tuning of retrieval-based models.

More critically, as the paper is mostly on theoretical analysis, the authors may want to improve the presentation and make sure that all the proofs are correct and self-explaining. For example:

1) Proof of Lemma B.2, on which the Theorem 3.4 is based, may not be correct. In the first inequality, the third term should be $8\delta L_l||\mathcal{F}^{loc}||_{\infty}$ instead of $4\delta L_l||\mathcal{F}^{loc}||_{\infty}$. In the second inequality, the authors should explain clearly why the expectation in the second term can be decomposed.
2) In Appendix C, ${R}_{{U}}^{\diamond}$ may need to be replaced with ${R}_{{U}, m}^{\diamond}$  for consistency.
3) The definition of $\hat{R}^{\text{ex}}_l(f)$ in Equation 37 seems to be wrong (should it be $\hat{R}^{\text{ex}}_l(f) = \frac{1}{n}\sum_{i\in [n]} l(f(x_i, \hat{D}^{x_i, r}), y_i)$?). There seems to be an unnecessary "that" in the following sentence. Such typos appeared at several places and the authors may want to carefully proofread the paper to improve the presentation.

**Summary Of The Paper:**

The authors proposed a theoretical framework to analyze the retrieval-based machine learning models for multi-class classification problems. The authors specifically focused on Local Empirical Risk Minimization~(Local ERM), in which the neighboring labeled data of a test sample is retrieved to train a model in a simple function class by minimizing the empirical risk given the retrieved data. The authors analyzed the expected excess risk of Local ERM compared to the global ERM, in which the empirical risk of all the training data was minimized to train a model in a more complex function class. The main results showed that under some assumptions, the expected excess risk of Local ERM is bounded by the sum of an approximation error term, which typically increases as the number of retrieved data increases, and a generalization error term, which typically decreases as the number of retrieved data increases.

The authors extended the analysis of Local ERM to a two-stage learning framework in which the first stage will learn a feature map whose mapping function can depend on all the training data. The authors also included the analysis of another retrieval-based framework, where the scorer jointly takes the test data and neighboring labeled data for prediction. Lastly, the authors experimentally showed that (I) There exists a trade-off between approximation error term and generalization error term in the Local ERM framework; (II) The Local ERM can achieve the result similar to the SOTA with a much simpler model.

**Summary Of The Review:**

The authors proposed a theoretical framework to analyze the retrieval-based machine learning models based on previous generalizability and learnability results. The presented theoretical and empirical results are interesting. The authors may need to check thoroughly their proofs to make sure that the presentation is correct and self-explaining as much as possible. It will be interesting if the authors can test some new learning strategies in retrieval-based models based on the analysis.

---

> ### Author Response · Authors · 2022-11-16
> **Response to Reviewer 94k1**
>
> We thank the reviewer for their constructive comments. We are glad that the reviewer acknowledged the timeliness of our study and found our theoretical analysis and empirical results interesting.
>
> As suggested by the reviewer, we plan to proofread the paper and ensure that the proofs are free of any typographical errors. Below, we have responded to the specific questions raised by the reviewer.
>
> > Proof of Lemma B.2, on which the Theorem 3.4 is based, may not be correct. In the first inequality, the third term should be $8\delta L\_l||\mathcal{F}^{loc}||\_{\infty}$ Instead of $4\delta L\_l||\mathcal{F}^{loc}||\_{\infty}$. In the second inequality, the authors should explain clearly why the expectation in the second term can be decomposed.
>
> The $4\delta L\_l||\mathcal{F}^{loc}||\_{\infty}$ term is correct. The sketch of the argument is as follows: We can expand the $g(X’,Y’) = \ell(\gamma\_f(X’,Y’)) - \ell(\gamma\_f(X,Y))$ and $g(x’,y’) = \ell(\gamma\_f(x’,y’)) - \ell(\gamma\_f(X,Y))$. This cancels out the common $\ell(\gamma\_f(X,Y))$  term. We then can bound the components $|\ell(\gamma_f(X’,Y’))|$  and $|\ell(\gamma\_f(x’,y’)) |$ each with $2 L\_{\ell} \|\mathcal{F}^{loc}\|\_{\infty}$. This gives the $4\delta L\_l||\mathcal{F}^{loc}||\_{\infty}$ bound.  We will add the derivation in the final version.
>
> > In Appendix C, ${R}_{{U}}^{\diamond}$ May need to be replaced with ${R}\_{{U}, m}^{\diamond}$ for consistency.
>
> Thanks for the suggestion. We will adopt the suggestion in the final version.
>
> > The definition of $\hat{R}_l^{ex}(f)$ in Equation 37 seems to be wrong (should it be $\hat{R}^{\text{ex}}\_l(f) = \frac{1}{n}\sum\_{i\in [n]} l(f(x\_i, \hat{D}^{x\_i, r}), y\_i)$?). There seems to be an unnecessary "that" in the following sentence. Such typos appeared at several places and the authors may want to carefully proofread the paper to improve the presentation.
>
> Thanks for catching these typographical errors. Yes, that is correct fix for Eq 37. We will carefully proofread the whole manuscript and correct the typographical errors, including the one mentioned above.

---

### Official Review · Reviewer_uQx9 · 2022-10-26

**Confidence:** 4
**Correctness:** 4
**Technical Novelty And Significance:** 2
**Empirical Novelty And Significance:** Not applicable
**Recommendation:** 3

**Clarity, Quality, Novelty And Reproducibility:**

This paper is well-organized and easy to follow for the reviewer.

The novelty is good as this paper is motivated by an interesting empirical question and attempts a theoretical formulation of the question.
- The studied approach seems quite related to nearest neighbor methods. Perhaps this should be discussed.

The appendix provides detailed proofs for the theoretical statements. I have not checked them in length but I also don't see any problem with the proofs.

**Strength And Weaknesses:**

Strength:
- This paper formulates a theoretical study of the generalization properties of retrieval based models.
- It also provides experiments to validate the generalization error bound.
- The experiments show that the performance of local ERM varies with the size of the neighborhood. In particular, it observes a trend in which the accuracy generally increases with the size of the retrieval set but then decreases afterwards.

Weakness:
- The significance of the contributed generalization bound is unclear; although understanding the generalization properties of retrieval-based models is a valuable quest, the current bounds rely on specific data regularity conditions and are generally difficult to interpret. Here're some specific feedback:
    - In eq (6), what does it mean that the complexity of $F^x$ is much less than the complexity of $F^global$?
    - In theorem 3.4, are all of the terms stated in the generalization bound properly defined? it would help provide a pointer to the def. of each term.
    - In theorem 4.1, you use $N(r, \delta)$ in the text but $N(r, \delta /n)$ in the equation. Is this due to a union bound over the training set?
- From a quick gloss of the appendix, the techniques also seem standard in the literature, so the technical contribution is limited at best.
    - It would also help if the main text could include some discussion about the technical contribution.
- It is also unclear whether and how tight are the proposed generalization bounds.
- The experiments are largely disconnected with the theoretical statements. There is a lack of elaboration about the benefits predicted by the theoretical results.

**Summary Of The Paper:**

This paper is motivated by the empirical success of retrieval-based classification models and therefore studies the generalization properties of these models. The models that this paper touches on broadly involve models which retrieves similar labeled examples from training data for an instance. The question that this paper seeks to answer is to provide a theoretical framework that can help analyze the retrieval set in ensuring the empirical performance of retrieval-based models.

The mathematical setup that this paper proposes is as follows:
- It considers a multi-class classification setting. There is a training set including i.i.d. samples.
- It assumes that the underlying data distribution follows a local-regularity structures; this ensures there exists a low-complexity function class that approximates the Bayes optimal for the local classification problem defined by these locally retrieved samples.

From a algorithm/model aspect, this paper focuses on two approaches:
- Local empirical risk minimization first retrieves a neighboring set; then, it identifies a local scorer/function by minimizing the empirical risk of the local set.

- Classification with extended feature space learns a common function over both the original instances and the retrieved neighboring labeled instances.

The main results of this paper provide generalization upper bounds for both of the above two approaches:
- The generalization bounds generally depend on some function that depends on the size of the neighborhood retrieval set (in the generalization errors of local ERM), the local and global optimal losses.
- A similar (albeit simpler) statement is stated for the second approach above.

**Summary Of The Review:**

Overall this paper provides a study of the generalization properties of retrieval-based models (in the sense of learning with several retrieved similar examples with a test instance). The main results involve generalization error bounds for two approaches (local ERM and extended feature space learning) under a local regularity condition. As stated above, the technical contribution is limited and the bounds could benefit from better interpretation and connection with empirical examples. Therefore, I'm recommending a "reject" for this submission.

---

> ### Author Response · Authors · 2022-11-16
> **Response to Reviewer uQx9 (Part 2)**
>
> > The studied approach seems quite related to nearest neighbor methods. Perhaps this should be discussed.
>
> Indeed, retrieval-based models are closely related to nearest-neighbor methods. In fact, we started our paper by stating that retrieval-based models can be viewed as combining instance-based models (such as nearest neighbor methods) with a parametric component. Some of the assumptions on the underlying data are also borrowed from the literature on k-NN models (e.g., Doring et al.). Furthermore, we also discuss k-NN models in our empirical evaluations. If the reviewer feels that further elaboration on the connection between retrieval-based models and nearest-neighbor methods will benefit the paper, we would be happy to oblige.
>
> [Doring et al.] Rate of convergence of k-nearest-neighbor classification rule, JMLR 2018.

---

> ### Author Response · Authors · 2022-11-16
> **Response to Reviewer uQx9 (Part 1)**
>
> We thank the reviewer for their detailed comments. Please see our point-by-point response to your questions/comments below. We hope that our responses will prompt the reviewer to reconsider their assessment of the paper. We are happy to engage in a follow-up discussion if more clarification is needed.
>
> > In eq (6), what does it mean that the complexity of Fx is much less than the complexity of Fglobal?
>
> Complexity here refers to the ‘size’ of the function classes. We can use Radamacher complexity as the measure of ‘size’ of a function class. This choice of ‘size’ will mean the Radamacher complexity of $F^{global}$ over the entire data distribution $D$ is much larger than the Radamacher complexity of $F^X$ over the data distribution $D^{X,r}$ for all $X$.
>
> > In theorem 3.4, are all of the terms stated in the generalization bound properly defined?
>
> In (I)  $\varepsilon\_X$ is defined in Eq. (6), $\varepsilon\_{\rm loc}$ is defined in Eq. (9). In (II)  the two $\mathcal{M}\_r(\cdot,\cdot,\cdot)$ terms are defined in Eq. (13). In (III) $\mathfrak{R}\_{\mathcal{R}^X}(\cdot)$ is the empirical Rademacher complexity defined in Definition A.1 in Appendix A (for lack of space). Will add these pointers as per your helpful suggestions.
>
> > In theorem 4.1, you use N(r,δ) in the text but N(r,δ/n) in the equation. Is this due to a union bound over the training set?
>
> Indeed, in Theorem 4.1 the $N(r, \delta /n)$ comes due to taking union bound over the data. We want to find a threshold $N$ such that probability $\mathbb{P}[ \mathcal{R}^{x\_i} \geq N, \forall i = 1, \dots, n ] \geq \delta$. Taking $N = N(r, \delta /n)$ satisfies the above. The probability needs to be satisfied over all data points as we need the empirically estimated extended feature to be close to the true extended feature  (distribution of the neighbors within $r$ radius).
>
> > From a quick gloss of the appendix, the techniques also seem standard in the literature, so the technical contribution is limited at best…
>
> Formalizing the theoretical framework for retrieval-augmented models is one of our main contributions. Unfortunately, even after the development of the framework, standard analysis techniques do not apply directly.
>
> For example, while analyzing local ERM in Section 3.1, the standard iid assumption does not hold as the retrieved set used to define the local-ERM objective depends on the current test instance. To circumvent this, we tactfully divide the excess risk into Global vs. Local components as detailed in Lemma 3.3. The next challenge is to capture the gains coming from performing ERM locally. Firstly, as described in a response above, restricting the ERM to lower complexity classes $\mathcal{F}^{\rm loc}$ reduces the generalization error. Furthermore, as we perform ERM in a locality of radius $r$, the variability of loss functions in local ERM becomes limited, and this leads to the reduction in generalization error that scales as $poly(r)$. Besides the challenges described above, Section 3.2 also requires us to deal with the data-dependent function class (cf.~(16)), where we successfully leverage the toolkit introduced by Foster et al.
>
> As for classification in the extended feature space (Section 4), we focus on learning approaches that directly map a test instance and associated retrieved set to a prediction, without explicitly solving an optimization problem like local ERM. Here, the main contribution lies in establishing a connection with the kernel methods so that one can leverage the powerful general framework for kernel-method in a novel setting.
>
> > It is also unclear whether and how tight are the proposed generalization bounds.
>
> As this is the *first attempt* at formalizing the performance of retrieval augmented training, we focus on studying the upper bounds for generalization. Deriving a lower bound for generalization to establish the tightness for our given setting is a very challenging task in general for most learning setup and left for future exploration in our setting.
>
> > The experiments are largely disconnected with the theoretical statements. There is a lack of elaboration about the benefits predicted by the theoretical results.
>
> We explain the impact of the retrieval set size on the model performance under *Implications of the excess risk bound* ( in page 6, above Section 3.2). Here we discuss how the error/accuracy curve suggested by Theorem 3.4 matches with the experimental results. In particular, the term (I) and (II) collectively capture the approximation error of a model, and both increase with increasing retrieval set size. The term (III) captures the generalization error which, for a fixed $\mathcal{F}^{\rm loc}$,  is shown to decrease with  increasing retrieval set size. This  shows up as the initial decrease in error with  increasing retrieval set size. Overall, accounting for the approximation and generalization terms, the total error has a U shaped curve with  increasing retrieval set size.

---

### Official Review · Reviewer_bKCw · 2022-10-26

**Confidence:** 1
**Clarity, Quality, Novelty And Reproducibility:** See summary
**Correctness:** 3
**Technical Novelty And Significance:** 3
**Empirical Novelty And Significance:** 2
**Recommendation:** 6

**Strength And Weaknesses:**

See summary

**Summary Of The Paper:**

This paper aims to provide theoretical characterizations of the generalization abilities of retrieval-based models. Specifically, this paper focuses on classification problems and examines two classes of retrieval-based approaches:

1. A local learning framework where a model is trained on retrieved examples and then used to make predictions for the test example.
2. A global model that treats retrieved examples as augmented features and directly make make predictions for the test example.

For the local learning framework, an empirical study is conducted to verify the theoretical results for two cases: one that uses the Euclidean distance for retrieval and the other employs a global learned neural representation. The performance of locally trained models improves as more examples are retrieved until a point where the local empirical risk minimization fails due to the lack of model capacity. This empirical study demonstrates the effectiveness of the local learning framework that allows small models to attain competitive performance.

Despite the complexity of theoretical analysis, the paper is presented in a way that the main points are clear and well-motivated. Since I am not an expert of the field, I was unable to verify the correctness of proofs; yet, the implication of the empirical study is straightforward and provides indirect support to authors’ argument.

**Summary Of The Review:**

See summary

---

> ### Author Response · Authors · 2022-11-16
> **Response to Reviewer bKCw**
>
> We thank the reviewer for taking the time to review our manuscript. We are glad that the reviewer found the presentation of the paper clear and well-motivated, and acknowledged the utility of our empirical study towards corroborating our theoretical analysis.

---

### Official Review · Reviewer_5HmN · 2022-10-28

**Confidence:** 3
**Correctness:** 2
**Technical Novelty And Significance:** 3
**Empirical Novelty And Significance:** 2
**Recommendation:** 5

**Clarity, Quality, Novelty And Reproducibility:**

Clarity:
The paper is written clearly in general, but some description and notation need to be clarified. For example, the (alpha, c)-weak margin condition, is this a reasonable assumption? What does this imply?

Notation: what does the \ perp mean in Assumption 3.2?

Quality:
The quality is fair given some missing pieces of both theoretical results and empirical evidence. See details above.

Novelty:
The problem studied is novel. The theoretical results seems to borrow some ideas from existing works, but in my opinion, the method is novel in general.

Reproducibility:
No code is provided, although I believe the results can be reproduced.

**Strength And Weaknesses:**

Strength:
1. Divide the retrieval-based models into two categories: a local empirical risk minimization problem, and a classification problem in an extended feature space, which provides more clear directions for empirical designs.
2. Detailed theoretical analysis on the local empirical risk minimization, which enable one to get better ideas on the underlying principles of the related retrieval-based models.
3. Empirical results also verify part of the theoretical results.

Weaknesses:
1. The focus of the paper is somewhat disconnected with practical settings. This paper gives more theoretical analysis on the local empirical risk minimization setting, which is rarely applicable in practice, as it needs to learn a separate classifier for each testing samples, which is very expensive. In practice, the extended feature space setting is usually adopted. Unfortunately, there is only limited theoretical results for this case. Moreover, there is no experiments corresponding to the provided results for this setting, i.e., the experiments only focus on the local empirical risk minimization setting. I actually think experiments for the extended feature space is necessary, as from the theoretical results, it indicates that the retrieval set size needs to scale at least logarithmically in the size of the training set, which is somewhat contradicted with practical setting, where typically a few (e.g,, 10) retrieval data points are used while still being able to get good results. I would expect some empirical evidence to verify this claim.
2. In the local empirical risk minimization setting, it is not clear how the retrieval set size impacts the generalization error. For the bound in Theorem 3.4, it seems to indicate that with larger retrieval set sizes, we can get smaller error. However, the empirical results indicates that larger retrieval set sizes can harm the performance if the classifier is too simple. I think this phenomenon is not reflected in the theoretical result. Is there a way to quantify this?

Also, do we know the increase rate of r in the (I) term in Theorem 3.4?

**Summary Of The Paper:**

This paper students the theoretical properties of two variants of retrieval-based models, a popular topic in improving large models. The two variants are formulated as a local empirical risk minimization problem and a classification problem in an extended feature space. Theoretical analysis are given in terms of the expected excess risk, with more detailed results for the local empirical risk minimization. Experiments are conducted to verify the theory for local empirical risk minimization.

**Summary Of The Review:**

This paper studies theoretical properties of retrieval based models from two perspectives. It develops expected excess risk for both cases, trying to show the factors impacting the risk, including the size of the retrieval set (which I believe is the most important one). The results are nice, but there are some discrepancies between the theoretical results and the empirical results, as well as some empirical practices.

---

> ### Author Response · Authors · 2022-11-16
> **Response to Reviewer 5HmN (Part 2)**
>
> > In the local empirical risk minimization setting, it is not clear how the retrieval set size impacts the generalization error…
>
> The model performance, excess risk to be precise, has two components: generalization error, and approximation error. We explain the impact of the retrieval set size, on these two terms under *Implications of the excess risk bound* ( in page 6, above Section 3.2). This also explains how the error/accuracy curve suggested by Theorem 3.4 matches with the experimental results.
>
> The term (I) and (II) collectively capture the approximation error and both increase with  increasing retrieval set size . We explain, for a fixed $\mathcal{F}^{\rm loc}$,  that term (I) increases with increasing retrieval set size (i.e. increase radius $r$) as the approximation over a larger set becomes harder.  For a fixed $\mathcal{F}^{\rm loc}$ the term (II) also increases with increasing retrieval set (polynomially with $r$) as the error of the center point $X$ diverges from the empirical error of the retrieved set of $\mathcal{R}^X$.   This explains why increasing retrieval set size increases the error as seen in the experiments.
>
> The term (III) captures the generalization error which,  for a fixed $\mathcal{F}^{\rm loc}$,  is shown to decrease with  increasing retrieval set size. This  shows up as the initial decrease in error with  increasing retrieval set size. Overall, accounting for the approximation and generalization terms, the total error has a U shaped curve with  increasing retrieval set size.
>
> > Also, do we know the increase rate of r in the (I) term in Theorem 3.4?
>
> The increase rate of the term (I) in r in our general setting is left as future work. The error is expected to scale as $poly(r)$, as seen in some special cases.
>
> - For a 1d $k+1$-times differentiable function class, namely $\mathcal{F}$, we can approximate this function class with the class of degree-$k$ polynomials, where the residual (L1 error) is $r^{(k+1)} \max_{f\in \mathcal{F}} | D^{(k+1)} f(X)  |$ for the data point $X$. Here $D^{(k+1)}$ denotes the $(k+1)$-th derivative of the function.
>
> - Multivariate polynomial approximations: While approximating a multivariate $d$-dimensional smooth functions with a quadratic polynomials the error scales as $O( (rd)^2 )$, while approximating with a linear function it scales as $O(rd)$  (can be inferred from Remark 6 in [Ruppert et al.]).
>
> [Ruppert et al.] Ruppert D, Wand MP. Multivariate locally weighted least squares regression. The annals of statistics. 1994 Sep 1:1346-70.
>
> We will present these quantitative examples in the final version of the paper.
>
> > … the (alpha, c)-weak margin condition ...
>
> The weak margin condition (Section 3) frequently appears in the literature and has been recently utilized in the analysis of k-NN classification [Doring et al.]. The weak margin condition implies that the probability that the margin of the scorer function is less than a threshold $t$ decays polynomially with $t$, i.e. $c t^{\alpha}$. Essentially, this imposes a specific structure on the dataset to be well-separable on most of the domain.
>
> [Doring et al.] Rate of convergence of k-nearest-neighbor classification rule, JMLR 2018.
>
> > What does the \ perp mean in Assumption 3.2?
>
> $A \perp B$ stands for independence of Random variables A and B. We will explain this in the final version.

---

> ### Author Response · Authors · 2022-11-16
> **Response to Reviewer 5HmN (Part 1)**
>
> Thank you for reviewing our paper. Below, we aim to individually address all of your comments/concerns.
>
> > The focus of the paper…
>
> The focus of the paper is establishing ‘a theoretical framework that can help rigorously showcase the value of the retrieved set in ensuring superior performance of modern retrieval-based models’, as mentioned in the introduction section (page 2). In this pursuit, we study two ways to explain the performance enhancements in the presence of retrieved sets: the models like transformers might be doing *implicit local learning* or operating on the *extended feature space*.  We are not advocating either of them to be applied as a practical method directly.
>
> > … Unfortunately, there is only limited theoretical results for this case…
>
> For the *extended feature space*,  we present excess risk bounds in Theorem 4.1 (similar to *local ERM*)  while delegating the bulk of technical content to Appendix D for space constraints. It is worth noting that despite being two very different approaches, final generalization bounds a very similar form: the sqrt(n) term, the logarithmic dependence of retrieved set size compared to training set size, etc.
>
> > … experiments for the extended feature space is necessary…
>
> Although our treatment of the extended feature space-based approach is a good theoretical **stepping stone** to study the general models, it is infeasible to run experiments with large datasets unlike the local ERM approach as Kernel methods do not scale well.
> That said, here we evaluate a kernel-based classifier over extended feature space for the synthetic dataset considered in Section 5. We design a kernel for extended feature space as mentioned in eq (40) in Appendix C. In particular, we use Gaussian-like function for $\kappa(\Psi(\mathsf{D}^{x\_1,r}), \Psi(\mathsf{D}^{x\_2,r})) = \exp(− \| \Psi(\mathsf{D}^{x\_1,r}) - \Psi(\mathsf{D}^{x\_1,r}) \|^2 /2\sigma^2\_{\kappa})$. To empirically estimate the distance between kernel mean embeddings of the two distributions we follow [Muandet17, Li15]. Also, $ k\_{\mathcal{X}}(x\_1, x\_2) = \exp(- \|x\_1-x\_2\|/2\sigma\_x^2 ) $ with normal L2 distance. Results from 1 run are tabulated below:
>
> | Num neighbors | Kernel machine |
> | --- | --- |
> | 2 | 0.776 |
> | 5 | 0.769 |
> | 10 | 0.777 |
> | 20 | 0.835 |
> | 50 | 0.819 |
> | 100 | 0.792 |
> | 200 | 0.585 |
>
> [Muandet17] Muandet, Krikamol, Kenji Fukumizu, Bharath Sriperumbudur, and Bernhard Schölkopf. "Kernel mean embedding of distributions: A review and beyond." Foundations and Trends in Machine Learning 10.1-2 (2017): 1-141.
>
> [Li15] Li, Yujia, Kevin Swersky, and Rich Zemel. "Generative moment matching networks." International conference on machine learning. PMLR, 2015.
>
> Note that, as the generalization bound suggested, that the model performance only improves up to a specific number of neighbors and starts degrading when further increasing the number of neighbors. (Also, see the next response regarding the exact scaling of the optimal size of neighborhood.)
>
> It’s also worth highlighting that a (4 layer) Transformer model that directly processes an instance along with the associated retrieved neighboring examples achieves a much higher performance of 0.898 with 10 neighbors. This is consistent with similar observations in the deep learning literature where kernel-based methods are often significantly outperformed by end-to-end neural networks. This is why, we highlight studying such end-to-end retrieval-based neural networks as an important future direction while concluding our paper (page 9).
>
> > …the retrieval set size needs to scale at least logarithmically in the size of the training set, which is somewhat contradicted with practical setting…
>
> Note that logarithmic scaling of retrieved sets means it will be very small, i.e., for a training set of size of few billions, its logarithm will be **~10**, as used in practice. The standard evaluation of GPT3 and PALM models are on 64-shot for most downstream tasks.

---

### Decision · Program_Chairs · 2023-01-20

**Decision:**

Reject

**Justification For Why Not Higher Score:**


Concerns raised by the reviewers include

"There are some discrepancies between the theoretical results and the empirical results"

"The significance of the contributed generalization bound is unclear"

"the technical contribution is limited and the bounds could benefit from better interpretation and connection with empirical examples. "

Please see the reviews for more details.

**Justification For Why Not Lower Score:**

N/A

**Metareview: Summary, Strengths And Weaknesses:**

Based on the discussion with the reviewers, it seems that the reviewers reached a consensus that the paper is not quite ready for publication at NeurIPS.

Concerns raised by the reviewers include

"There are some discrepancies between the theoretical results and the empirical results"

"The significance of the contributed generalization bound is unclear"

"the technical contribution is limited and the bounds could benefit from better interpretation and connection with empirical examples. "

Please see the reviews for more details.